# The N-terminus of the prion protein is a toxic effector regulated by the C-terminus

Bei Wu[1], Alex J McDonald[1], Kathleen Markham[2], Celeste B Rich[1],
Kyle P McHugh[3], Jörg Tatzelt[4,5], David W Colby[3], Glenn L Millhauser[2],
David A Harris[1]*

[1]Department of Biochemistry, Boston University School of Medicine, Boston, United States; [2]Department of Chemistry and Biochemistry, University of California, Santa Cruz, United States; [3]Department of Chemical and Biomolecular Engineering, University of Delaware, Newark, United States; [4]Department of Biochemistry of Neurodegenerative Diseases, Institute of Biochemistry and Pathobiochemistry, Ruhr University Bochum, Bochum, Germany; [5]Neurobiochemistry, Adolf Butenandt Institute, Ludwig Maximilians University, Munich, Germany

**Abstract** PrP$^C$, the cellular isoform of the prion protein, serves to transduce the neurotoxic effects of PrP$^{Sc}$, the infectious isoform, but how this occurs is mysterious. Here, using a combination of electrophysiological, cellular, and biophysical techniques, we show that the flexible, N-terminal domain of PrP$^C$ functions as a powerful toxicity-transducing effector whose activity is tightly regulated *in cis* by the globular C-terminal domain. Ligands binding to the N-terminal domain abolish the spontaneous ionic currents associated with neurotoxic mutants of PrP, and the isolated N-terminal domain induces currents when expressed in the absence of the C-terminal domain. Anti-PrP antibodies targeting epitopes in the C-terminal domain induce currents, and cause degeneration of dendrites on murine hippocampal neurons, effects that entirely dependent on the effector function of the N-terminus. NMR experiments demonstrate intramolecular docking between N- and C-terminal domains of PrP$^C$, revealing a novel auto-inhibitory mechanism that regulates the functional activity of PrP$^C$.

*For correspondence: daharris@ bu.edu

**Competing interests:** The authors declare that no competing interests exist.

## Introduction

Prion diseases, or transmissible spongiform encephalopathies, comprise a group of fatal neurodegenerative disorders in humans and animals for which there are no effective treatments or cures. These diseases are caused by refolding of the cellular prion protein (PrP$^C$) into an infectious isoform (PrP$^{Sc}$) that catalytically templates its abnormal conformation onto additional molecules of PrP$^C$ (*Prusiner, 1998*). A similar, prion-like process may play a role in other neurodegenerative disorders, such as Alzheimer's and Parkinson's diseases and tauopathies, which are due to protein misfolding and aggregation (*Jucker and Walker, 2013*).

There is evidence that PrP$^C$, in addition to serving as a precursor to PrP$^{Sc}$, acts as a signal transducer that mediates the neurotoxic effects of PrP$^{Sc}$ (*Biasini et al., 2012*; *Brandner et al., 1996*; *Chesebro et al., 2005*; *Mallucci et al., 2003*). Clues to possible mechanisms by which PrP$^C$ can initiate neurotoxic activity have emerged from studies of transgenic mice expressing PrP molecules that harbor certain internal deletions within the N-terminal domain. The PrP$^C$ molecule consists of a partially unstructured N-terminal domain (residues 23–125), and a globular, C-terminal domain (residues 126–230) comprising three α-helices and two short, β-strands (*Zahn et al., 2000*). Deletions spanning a 21-amino acid region (amino acids 105–125) at the end of the flexible, N-terminal domain induce a spontaneous neurodegenerative phenotype with certain similarities to natural prion

**eLife digest** Prion diseases are a group of degenerative illnesses of the brain caused when a molecule called the prion protein (PrP for short) adopts the wrong shape. These diseases include the human form of mad cow disease, and are often fatal with no effective treatments or cures. Though the normal activity of PrP is not certain, abnormal PrP can affect the healthy PrP on the surface of brain cells and lead to disease. Similar mechanisms may also contribute to other life-threatening brain disorders, including Alzheimer's disease and Parkinson's disease.

It had been shown that certain altered PrP proteins caused the death of brain cells by allowing excessive electrical charges to cross the membranes of the cell. These changes led to symptoms in animal models of the diseases. Experiments showed that adding a large amount of normal PrP to the cells could prevent these effects. These studies, however, had not yet resolved how PrP behaves inside cells and how this contributes to disease.

Using genetically modified mice and cells grown in the laboratory, Wu et al. investigated the role of different parts of PrP in causing brain cells to degenerate. The experiments showed that one end of the protein, called the N-terminus, is involved in the movement of electrical charges across the cell membrane and is able to cause cell degeneration. By contrast, the other end of the protein, the C-terminus, acts as a regulator for the N-terminus and can prevent cell degeneration. Further investigation revealed that the C-terminus regulates the N-terminus through direct contact.

A better understanding of the role of PrP in prion diseases may help to reveal new treatments for these and other degenerative brain disorders. In particular, the new findings highlight that treatments should target the toxic N-terminus of altered PrP and not the regulatory C-terminus. Further study will examine how different molecules in the brain control the interaction between the two ends of PrP in healthy brain cells and how this is altered in diseased cells.

diseases, but without accumulation of PrP$^{Sc}$ (*Baumann et al., 2007*; *Li et al., 2007*; *Shmerling et al., 1998*). Importantly, these phenotypes are dose-dependently suppressed by co-expression of wild-type PrP, suggesting that the wild-type and deleted molecules interact with each other, or compete for binding to a common molecular target that mediates both physiological and pathological effects. The shortest deletion, Δ105–125 (designated ΔCR, for central region), produces the most severe neurodegenerative phenotype, and requires the largest amount of wild-type PrP for rescue (*Li et al., 2007*).

In our efforts to understand why these deleted forms of PrP are so neurotoxic, we have discovered that they induce large, spontaneous ionic currents, recordable by patch clamping techniques, when expressed in a variety of cell lines (*Solomon et al., 2010*, *2011*) and in primary neurons (*Biasini et al., 2013*). Remarkably, these currents are silenced by co-expression of wild-type PrP in the same cells, paralleling the rescuing effects of wild-type PrP in transgenic mice expressing deleted PrP. This observation suggests that the spontaneous ionic currents themselves, or some closely associated phenomenon, play a role in the neurodegenerative phenotype of these mice.

In this study, we uncover novel mechanistic features of the toxicity-inducing activities of PrP$^C$. We show that ligands binding N-terminal domain of PrP$^C$ abolish ΔCR PrP-induced currents, as do mutations of positively charged residues at the extreme N-terminus of this domain. Remarkably, expression of the isolated N-terminal domain in the absence of the C-terminal domain also induces spontaneous currents, indicating that the N-terminal domain is capable of acting as an autonomous, toxicity-determining effector. We also demonstrate that anti-PrP antibodies targeting epitopes in the structured, C-terminal domain induce ionic currents in cultured cells expressing wild-type PrP$^C$, and cause degeneration of dendrites on hippocampal neurons. These results, taken together with structural evidence from heteronuclear NMR experiments, suggest a molecular model for PrP$^C$ in which the N-terminal domain acts as a neurotoxic effector whose activity is regulated by the C-terminal domain. We speculate that this inter-domain regulatory interaction could play a role in the physiological function of PrP$^C$, and that disruption of this interaction could contribute to pathology in neurodegenerative disorders. Our results also have important implications for the safety of anti-PrP antibody therapies for prion and Alzheimer's diseases.

## Results

### N-terminal ligands, and reversal of positive charges, block ΔCR-induced currents

Our previous studies identified a positively charged, nine amino acid segment at the very beginning of the N-terminal domain (residues 23–31, KKRPKPGGW) that is essential for the current activity of ΔCR PrP, and for the neurodegenerative phenotype of mice expressing another deletion mutant, Δ32–134 (*Solomon et al., 2011*; *Westergard et al., 2011b*). As demonstrated in these earlier studies, deletion of residues 23–31 abolishes the spontaneous inward currents induced by ΔCR PrP (*Figure 1—figure supplement 1A*), as does addition of pentosan polysulfate (PPS), a negatively-charged glycosaminoglycan which binds to several regions within the N-terminal domain (*Figure 1—figure supplement 1B*). To probe further the role of the N-terminal domain in ΔCR PrP-induced currents, we tested the effect of additional ligands (antibodies and $Cu^{2+}$ ions), as well alteration of positively charged residues within 23–31 region.

We tested two different anti-prion antibodies: 100B3 (*Thuring et al., 2005*), targeting residues 24–28, and POM11 (*Polymenidou et al., 2008*), which binds to the octapeptide repeats (residues 51–90). These antibodies (at concentrations of 57 nM and 33 nM, respectively), dramatically reduced ΔCR PrP-induced currents (*Figure 1A,B*). As a control for POM11, we tested the effect of this antibody on ΔCR/Δ51–90 PrP, which retains current activity (*Figure 1—figure supplement 1*) as shown previously (*Solomon et al., 2011*), but which lacks the octapeptide repeat region and would not be expected to bind the antibody. As predicted, POM11 had relatively little effect on the currents induced by ΔCR/Δ51–90 PrP (*Figure 1B*), while 100B3 and PPS were still inhibitory (*Figure 1A*, *Figure 1—figure supplement 1B*). These results demonstrate that antibody ligands targeting the extreme N-terminus or the octapeptide repeats inhibit ΔCR PrP-induced currents. Interestingly, even though the octapeptide repeats are not required for current activity, binding of an antibody ligand to this region blocks the currents.

To further investigate the role of the octapeptide repeats, we tested the effect of $Cu^{2+}$ ions, which are known to bind to histidine residues located within four of the five octapeptide repeats and at positions 95 and 110 (*Millhauser, 2007*). We found that treatment with $Cu^{2+}$-pentaglycine (100 μM) blocked spontaneous currents in N2a cells expressing ΔCR PrP, but not in cells expressing ΔCR/Δ51–90 (*Figure 1C*). Pentaglycine, with a $K_d$ for $Cu^{2+}$ similar to that of $PrP^C$ (40 nM), was included as a $Cu^{2+}$ chelator in order to minimize the concentration of free $Cu^{2+}$ (1 μM) while still allowing $PrP^C$ to compete for $Cu^{2+}$ ions. This result indicates that $Cu^{2+}$ coordination to the octapeptide repeats blocks ΔCR PrP currents.

Both PPS and copper have been shown to induce endocytosis of $PrP^C$ from cell surface (*Brown and Harris, 2003*; *Pauly and Harris, 1998*; *Shyng et al., 1995a*). To determine whether ligand binding to the N-terminus blocked ΔCR PrP-induced currents by reducing the amount of the mutant protein on the cell surface, living cells were treated with PPS for 1 hr or $Cu^{2+}$-pentaglycine for 5 min in recording buffer at room temperature, and then surface-stained for ΔCR PrP with anti-PrP antibody. Under these conditions, neither ligand altered the amount of ΔCR PrP on the cell surface (*Figure 1—figure supplement 2*), presumably because of the short period of treatment and the fact that the experiment was conducted at room temperature rather than 37°C. As reported previously (*Brown and Harris, 2003*; *Pauly and Harris, 1998*; *Shyng et al., 1995a*), PPS did induce significant endocytosis after treatment at 37°C for 48 hr (*Figure 1—figure supplement 2*). Cells treated with 100B3 or POM11 for 48 hr at 37°C did not show any change in surface level of ΔCR PrP (*Figure 1—figure supplement 2*). Thus, the inhibitory effects of PPS, POM11, 100B3 and $Cu^{2+}$ did not result from acute changes in localization or trafficking of the mutant protein.

The extreme N-terminus of $PrP^C$ contains four positively charged amino acids ($_{23}$**KKR**P**K**PGGW$_{31}$). To test the role of these residues in ΔCR-induced currents, we generated a variant of ΔCR (designated ΔCR/E3D) in which the three lysine residues (at positions 23, 24, and 27) were mutated to glutamic acid and the single arginine residue (at position 25) was mutated to aspartic acid. We found that N2a cells expressing this mutant did not show any currents (*Figure 1—figure supplement 1*). This result indicates that the positive charges within the 23–31 segment are essential for ΔCR-induced spontaneous currents.

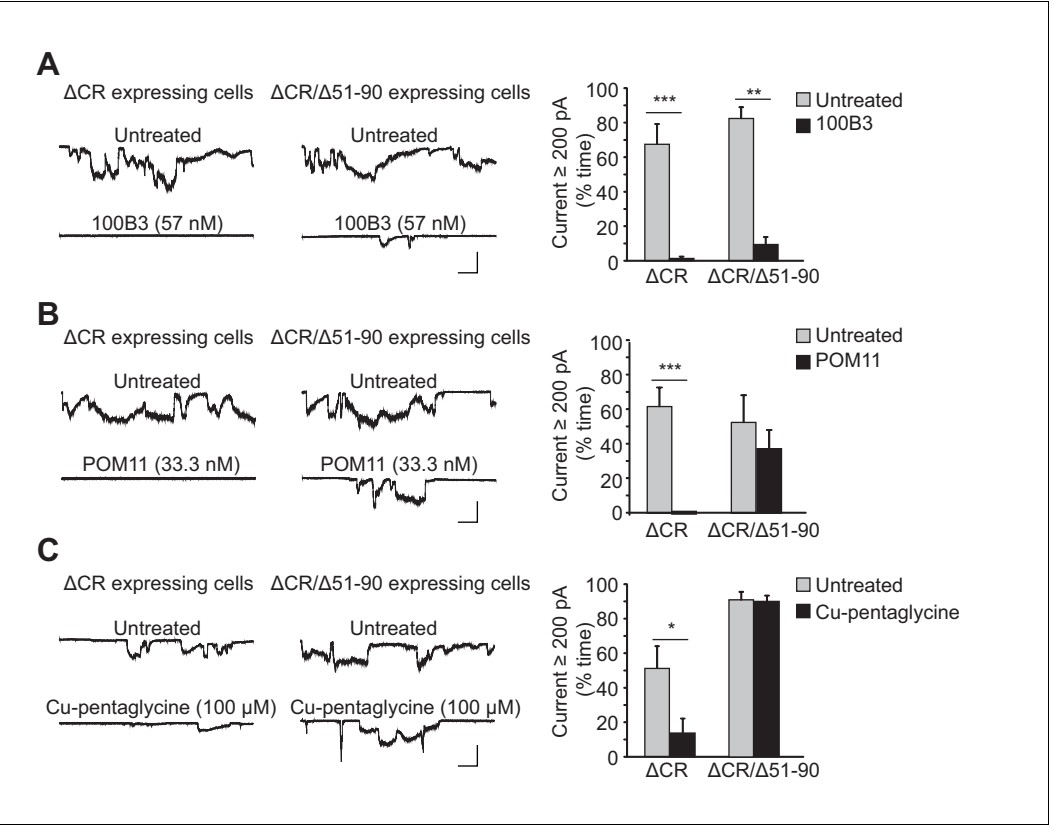

**Figure 1.** Ligands binding to the N-terminal domain of PrP[C] block ΔCR-induced currents. (**A**) Left, representative traces of currents recorded from N2a cells expressing ΔCR or ΔCR/Δ51–90 PrP in the absence (upper traces) or presence (lower traces) of 100B3 (57 nM). Right, quantitation of the currents, plotted as the percentage of the total time the cells exhibited inward current ≥200 pA (mean ± S.E.M., n = 10). (**B**) Left, representative traces of currents recorded from N2a cells expressing ΔCR or ΔCR/Δ51–90 PrP in the absence (upper traces) or presence (lower traces) of POM11 (33.3 nM). Right, quantitation of the currents (mean ± S.E.M., n = 10). (**C**) Left, representative traces of currents recorded from N2a cells expressing ΔCR or ΔCR/Δ51–90 PrP in the absence (upper traces) or presence (lower traces) of Cu-pentaglycine (100 µM). Right, quantitation of the currents (mean ± S.E.M., n = 10). Scale bars in all panels: 1 nA, 30 s. *p<0.05; **p<0.01; ***p<0.005.

The following source data and figure supplements are available for figure 1:

**Source data 1.** Quantification of ΔCR PrP-induced currents w/o treatment of ligands binding to PrP[C] N-terminus.

**Figure supplement 1.** Mutated forms of PrP induce spontaneous currents.

**Figure supplement 2.** Surface immunofluorescence staining of PrP[C] on N2a cells expressing ΔCR after treatment with N-terminal ligands.

## The N-terminal domain of PrP[C] induces ionic currents in the absence of the C-terminal domain

Having shown that the N-terminal domain of PrP[C] is essential for ΔCR PrP-induced currents, we wished to determine whether N-terminus is, by itself, sufficient to induce spontaneous currents. We constructed a series of chimeric proteins (collectively designated PrP(N)-EGFP-GPI) consisting of various lengths of the N-terminal domain of PrP[C] (residues 23–109) fused to an EGFP molecule that was equipped with the GPI addition signal from PrP[C] (*Figure 2A*). It was necessary to include the EGFP moiety to enable efficient delivery of the protein to the cell surface (*Heske et al., 2004*); fusing the N-terminal domain directly to the GPI addition signal results in a protein that is largely retained in the ER and is degraded by the proteasome (*Dametto et al., 2015*). We confirmed cell surface

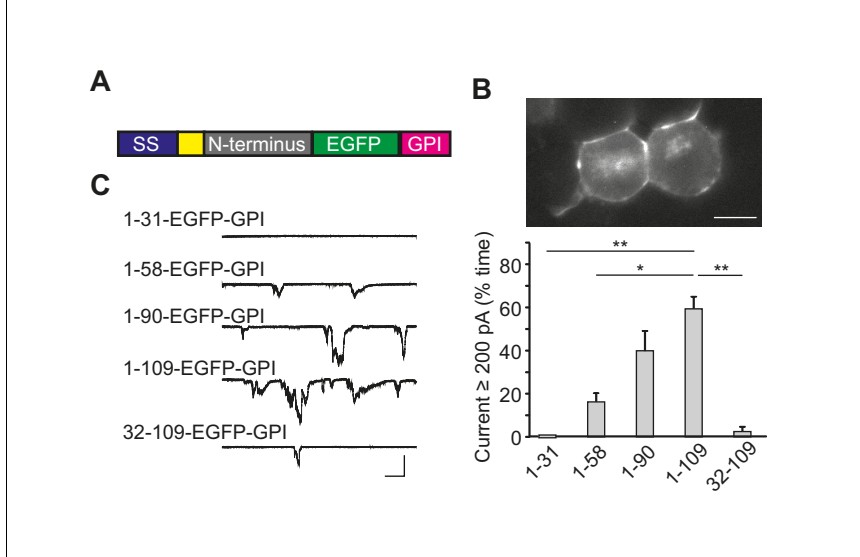

**Figure 2.** The N-terminal domain of PrP$^C$ induces ionic currents in the absence of the C-terminal domain. (**A**) Schematic of PrP(N)-EGFP-GPI constructs containing the N-terminus of PrP fused to EGFP and the PrP GPI attachment sequence. The colored blocks represent the signal sequence (blue), polybasic residues 23–31 (yellow), different portions of the N-terminus (grey), EGFP (green), and the GPI attachment sequence (magenta). (**B**) Fluorescence image of N2a cells expressing PrP(1-109)-EGFP-GPI, showing localization of the protein on the cell surface. Scale bar = 10 μm. (**C**) Left, representative traces of currents recorded from N2a cells expressing constructs with different lengths of the N-terminus (1–31, 1–58, 1–90, 1–109 and 32–109). Scale bars: 500 pA, 30 s. Right, quantitation of the currents, plotted as the percentage of the total time the cells exhibited inward current ≥200 pA (mean ± S.E.M., n = 5 cells). *p<0.05; **p<0.01.

The following source data and figure supplement are available for figure 2:

**Source data 1.** Quantification of N1-GFP-GPI-induced currents N2a cells.

**Figure supplement 1.** Currents induced by PrP(1-109)-EGFP-GPI have the same features as currents induced by ΔCR PrP.

---

localization of the PrP(N)-EGFP-GPI constructs in transfected cells by fluorescence microscopy (**Figure 2B**).

We found that cells expressing PrP(N)-EGFP-GPI constructs displayed spontaneous ionic currents. The most active currents were observed with a construct encompassing PrP residues 1–109, with successively lower current activities seen as the constructs became shorter (**Figure 2C,D**). Importantly, the PrP 32–109 construct was much less active, demonstrating the dependence of current activity on the 23–31 region (**Figure 2C,D**). Expression of the C-terminal domain of PrP$^C$ (Δ23–89 PrP) did not induce any currents (**Figure 2—figure supplement 1A**).

We note that ΔCR/Δ51–90 PrP (**Figure 1** and **Figure 1—figure supplement 1**) was more effective at inducing currents than 1–31-EGFP or 1–59-EGFP (**Figure 2**). There are at least two possible reasons for this observation. First, the ΔCR/Δ51–90 PrP construct contains additional sequences that are not present in the 1–31-EGFP or 1–59-EGFP constructs. In particular, ΔCR/Δ51–90 PrP contains residues 91–104, which are absent in 1–31-EGFP and 1–59-EGFP. These additional residues may enhance production of spontaneous currents, consistent with the general observation that PrP-EGFP chimeras incorporating longer stretches of the PrP N-terminus produced more currents (**Figure 2C**). A second possible explanation is that the EGFP portion of the chimeric constructs may position the PrP N-terminus at a different distance from the membrane, or in a different orientation, than the natural PrP$^C$ C-terminus, and this may diminish the ability of the N-terminus to interact with the membrane to produce currents.

The spontaneous currents associated with PrP(N)-EGFP-GPI had characteristics identical to the currents associated with ΔCR PrP. First, PrP(N)-EGFP-GPI currents were sporadic in nature, and were silenced by N-terminal ligands, including PPS, 100B3, POM11, and $Cu^{2+}$-pentaglycine (*Figure 2—figure supplement 1B*). Second, PrP(N)-EGFP-GPI currents, like ΔCR PrP currents, were silenced in a dose-dependent fashion by co-expression of WT PrP (*Figure 2—figure supplement 1C and D*). Moreover, removal of 23–31 region abolished the ability of WT PrP$^C$ to suppress both ΔCR and PrP (N)-EGFP-GPI currents (*Figure 2—figure supplement 1E and F*), which is consistent with the observation that expression of Δ23–31 PrP does not rescue the neurodegenerative phenotype of mice expressing Δ32–134 PrP (*Turnbaugh et al., 2011*). Finally, the currents induced by both ΔCR and PrP(N)-EGFP-GPI were voltage-dependent, only being observed at holding potentials below −30 mV (*Figure 2—figure supplement 1G and H*).

## NMR analysis reveals diminished interaction between the N- and C-terminal domains of ΔCR PrP

The results presented thus far suggest that the C-terminal domain of PrP$^C$ may directly regulate the N-terminal domain through a cis-interaction between the two domains. This interaction may be disrupted by deletion of residues in the central region (as in ΔCR), or by substitution of an unrelated protein for the C-terminal domain (as in PrP(N)-EGFP-GPI). Recently, *Evans et al. (2016)* used $^1H$-$^{15}N$ HSQC NMR to demonstrate that $Cu^{2+}$ ions, when bound to the N-terminal, octapeptide repeats, promote contact between these repeats and a C-terminal surface site encompassing helices 2 and 3. We predicted that this *cis* interaction would be weakened in ΔCR, thereby accounting for the ability of the liberated N-terminal domain to induce spontaneous currents.

To test this prediction, we employed paramagnetic relaxation enhancement to probe the interaction between octapeptide-bound $Cu^{2+}$ ions and residues in the C-terminal domain. Using the methods of *Evans et al. (2016)*, we compared the $^1H$-$^{15}N$ HSQC NMR spectra of recombinant, wild-type mouse PrP (WT PrP) and ΔCR PrP in the presence and absence of one equivalent of $Cu^{2+}$ (*Figure 3—figure supplement 1*). Residues that come in close proximity to $Cu^{2+}$ have their NMR cross-peaks broadened, resulting in a lower observed intensity. For example, in the absence of $Cu^{2+}$, a large cross-peak shown in black corresponding to WT PrP residue E199 was observed (*Figure 3—figure supplement 1–A1*). However, upon the addition of one equivalent of $Cu^{2+}$ the corresponding cross-peak depicted in red for residue E199 was greatly diminished in intensity. On the other hand, when $Cu^{2+}$ was titrated into ΔCR PrP, only a small shift between positions of the cross-peaks corresponding to E199 was observed (*Figure 3—figure supplement 1–A2*). The chemical shift changes, mapped onto the structure of WTPrP in *Figure 3A,B*, identify those residues in the C-terminal domain affected by $Cu^{2+}$ binding to the octapeptide repeats. In *Figure 3A1 and A2*, residues that do not significantly change upon addition of $Cu^{2+}$ are indicated by blue bars, while residues that underwent a significant reduction in intensity are indicated by red bars. Cross-peaks that were not identified are not shown in the figure. The large interaction patch seen in WT PrP is clearly reduced in the ΔCR mutant (compare the residues highlighted in magenta in *Figure 3B1/C1 and B2/C2*), especially in helices 2 and 3. These data suggest that deletion of residues in the central region disrupts a $Cu^{2+}$-driven regulatory interaction between the N- and C-terminal domains.

## Antibodies against the C-terminal domain and hinge region of PrP$^C$ induce ionic currents

Previous studies have reported that antibodies targeting specific epitopes in the structured domain of PrP$^C$ cause neuronal death when administered in vivo or in brain slices (*Reimann et al., 2016*; *Solforosi et al., 2004*; *Sonati et al., 2013*). We wondered whether the neurotoxicity of anti-PrP antibodies might be due to their ability to induce ionic currents, similar to the way that ΔCR PrP causes ionic currents in cultured cells (*Solomon et al., 2010*, *2011*) and neuronal death in transgenic mice (*Li et al., 2007*). We found that two antibodies targeting overlapping epitopes encompassing helix 1 in the C-terminal half of PrP$^C$, POM1 (*Polymenidou et al., 2008*; *Sonati et al., 2013*) and D18 (*Doolan and Colby, 2015*; *Williamson et al., 1998*), induced spontaneous currents in N2a cells expressing WT PrP$^C$ (*Figure 4A*).

A third anti-prion antibody with a similar epitope, ICSM-18 (*Antonyuk et al., 2009*), had a comparable effect, which was blocked by PPS and was absent with non-specific mouse IgG (*Figure 4—*

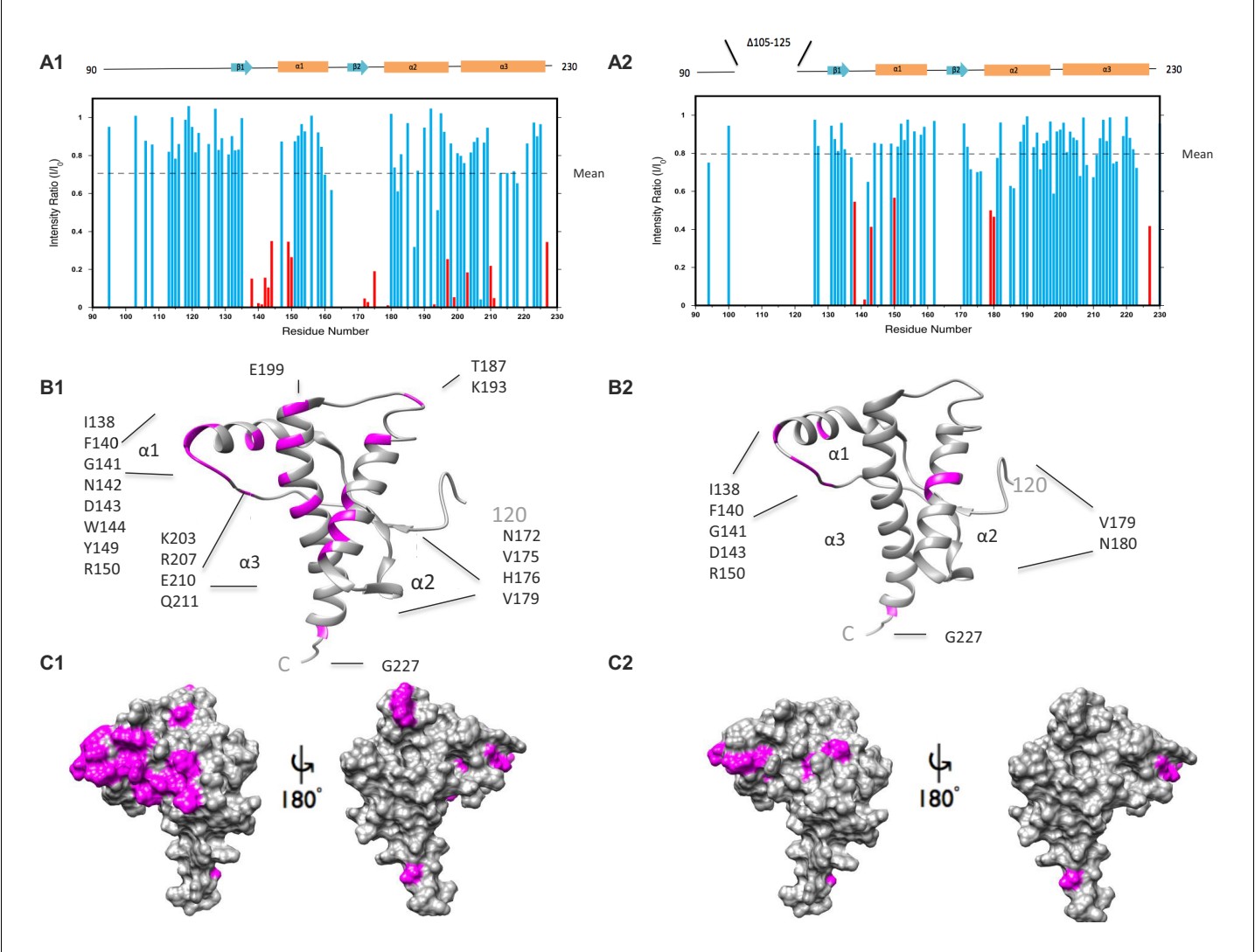

**Figure 3.** ΔCR PrP shows diminished interaction between N- and C-terminal domains based on NMR analysis. (**A**) Reduction in peak intensities of $^1$H-$^{15}$N HSQC spectra of WT PrP (A1) and ΔCR PrP (A2) in the presence of $Cu^{2+}$. Data are shown only for the structured domain (residues 90–230). $I/I_0$ values represent the ratios of peak heights in the presence and absence of 1 equivalent of $Cu^{2+}$. Residues with $I/I_0$ values less than 1.0 SD below the mean (dotted line) are shown in red, with unassigned residues omitted. (**B**) Residues (labeled and colored magenta) of WT PrP (B1) and ΔCR PrP (B2) with $I/I_0$ values < 1.0 SD below the mean, mapped onto a ribbon representation of the NMR structure of mouse PrP(120-230) (PDB:1XYX). (**C**) Affected residues (magenta) of WT PrP (C1) and ΔCR PrP (C2) are mapped onto surface plots of mouse PrP(120-230) (PDB:1XYX).

The following figure supplement is available for figure 3:

**Figure supplement 1.** NMR signals for C-terminal residues broaden in the presence of $Cu^{2+}$ bound to the N-terminal octarepeats.

*figure supplement 1A*). Because we were not able to obtain sufficient amounts of ICSM-18 for further electrophysiological experiments, we turned to a single chain version of this antibody (ICSM-18 scFv). Like the holo-antibody, ICSM-18 scFv-induced spontaneous currents on on N2a cells overexpressing WT PrP$^C$ (*Figure 4—figure supplement 1B*), although higher concentrations were required (200 nM for the scFv version, compared to 33.3 nM for the holo-antibody), presumably reflecting the lower avidity of the monovalent scFv antibodies (*Mammen, 1998*). The spontaneous currents induced by ICSM-18 scFv were abolished by PPS (100 µg/ml), and were absent with a control scFv (*Figure 4—figure supplement 1B*).

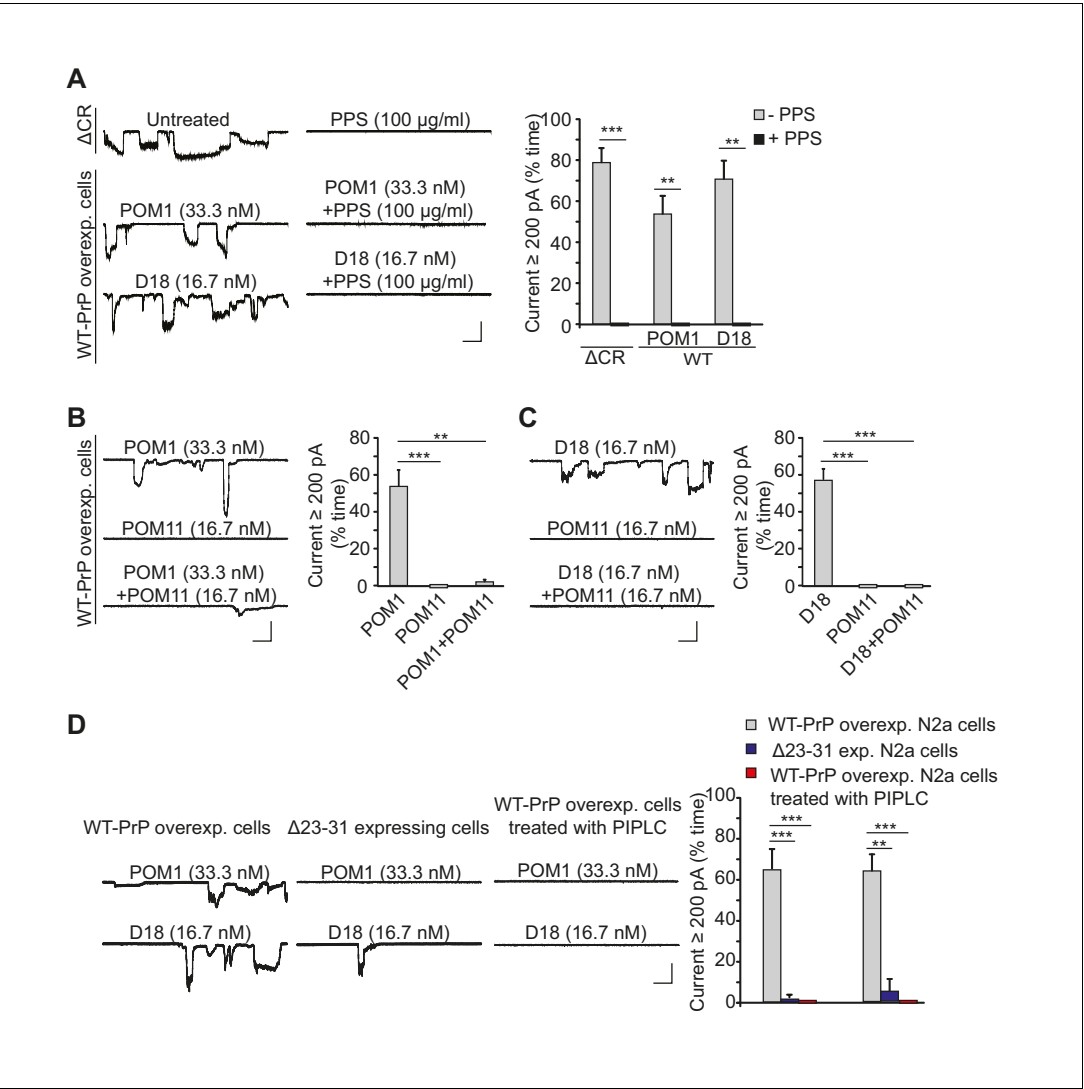

**Figure 4.** Antibodies against the C-terminal domain induce ionic currents in N2a cells expressing wild-type PrP[C]. (A) Left, representative traces of spontaneous currents recorded from cells expressing ΔCR PrP (top traces), and currents induced by anti-prion antibodies (POM1 and D18) in cells expressing WT PrP (lower traces) in the absence (left-hand traces) or presence (right-hand traces) of PPS (100 ug/ml). Right, quantitative analysis of the currents, plotted as the percentage of the recording time the cells exhibited inward currents ≥200 pA. (mean ± S.E.M., n = 10). (B) Left, representative traces of currents recorded from cells expressing WT PrP in the presence of POM1, POM11, or POM1+POM11. Right, quantitative analysis of the currents (mean ± S.E.M., n = 10). (C) Left, representative traces of currents recorded from cells expressing WT PrP in the presence of D18, POM11, or D18 +POM11. Right, quantitative analysis of the currents (mean ± S.E.M., n = 10). (D) Left, representative traces of currents induced by POM1 (upper traces) or D18 (lower traces) in cells expressing WT PrP (left-hand traces) or Δ23–31 PrP (center traces), or in cells expressing WT PrP after pretreatment with PIPLC (1.0 units/ml for 4 hr at 37°C) (right-hand traces). Right, quantitative analysis of the currents (mean ± S.E.M., n = 10). Scale bars in all panels: 1 nA, 30 s. **p<0.01; ***p<0.005.

The following source data and figure supplements are available for figure 4:

**Source data 1.** Quantification of anti-prion antibody-induced currents on N2a cells.
**Figure supplement 1.** ICSM-18 induces currents in N2a cells.
**Figure supplement 2.** Antibody-induced currents are dependent on PrP[C] expression level and are produced by Fab fragments.

*Figure 4 continued on next page*

*Figure 4 continued*

**Figure supplement 3.** 6D11, but not other central region antibodies, weakly induces currents in N2a cells.

**Figure supplement 4.** POM1, but not antibodies recognizing other regions of the C-terminal domain, induces currents.

**Figure supplement 5.** D18 induces currents in Tga20 hippocampal neurons over-expressing WT PrP and in wild-type neurons, but not in PrP knock-out (KO) neurons.

The properties of the D18-, POM1-, and ICSM-18-induced currents were identical to those of the spontaneous currents associated with ΔCR PrP, in terms of their sporadic nature, and suppression by PPS (*Figure 4A*, *Figure 4—figure supplement 1*). In addition, treatment of WT PrP-expressing N2a cells with POM11 blocked POM1- or D18-induced currents (*Figure 4B and C*), similar to the inhibitory effect of POM11 on ΔCR PrP currents (*Figure 1*). Finally, cells expressing PrP with a deletion of residues 23–31 were resistant to current induction by POM1 and D18 antibodies (*Figure 4D*), parallel to the situation with ΔCR PrP, and emphasizing the importance of the polybasic region in antibody-induced current generation. One would expect a significant reduction in antibody-induced currents in N2a cells expressing Δ23–31 PrP compared to cells over-expressing WT PrP (as observed in *Figure 4D*), since the endogenous level of WT PrP in N2a cells is low, and we have shown that untransfected cells display reduced currents after antibody treatment (*Figure 4—figure supplement 2A*). The fact that the Δ23–31-expressing cells do not display even low levels of current due to endogenous $PrP^C$ may be due to some suppression of endogenous PrP expression, or perhaps competition by the deleted protein for binding of the antibodies.

We performed several control experiments to demonstrate that the antibody-induced currents were dependent on expression of cell-surface $PrP^C$. First, we pretreated transfected N2a cells expressing WT PrP with phosphatidylinositol-specific phospholipase C (PIPLC) at 1 U/ml for 4 hr at 37°C, which cleaves the C-terminal GPI anchor and releases $PrP^C$ from the cell membrane. Neither D18 nor POM1 induced currents in PIPLC pretreated cells (*Figure 4D*). Second, we demonstrated that D18 did not induce currents in N2a cells in which PrP gene expression had been abolished by CRISPR-Cas technology (*Mehrabian et al., 2014*) (*Figure 4—figure supplement 2A*). Finally, we observed that D18 induced currents in untransfected N2a cells, although they were smaller than in transfected N2a cells over-expressing WT $PrP^C$, presumably due to the lower expression level of $PrP^C$ in the former cells (*Figure 4—figure supplement 2A*). Taken together, these results indicate that D18 and POM1-induced currents are PrP-dependent and are due to binding of the antibodies to cell-surface $PrP^C$.

To determine whether antibody-induced currents were the result of cross-linking of cell-surface $PrP^C$, or Fc-mediated antibody effector functions, we tested the effect of monovalent Fab fragments. We found that Fab fragments prepared from D18-induced currents on N2a cells (*Figure 4—figure supplement 2B*). This result, along with the effectiveness of ICSM-18 scFv antibody (*Figure 4—figure supplement 1B*), indicates that the ability of the C-terminally directed antibodies to induce currents in PrP-expressing cells is not due to cross-linking of cell-surface PrP by bivalent binding, or to Fc-mediated effector functions such as complement fixation.

The antibody 6D11 (*Pankiewicz et al., 2006*), whose epitope (residues 93–109) encompasses several positively charged residues following the octapeptide repeats, also induced currents in WT PrP-expressing N2a cells (*Figure 4—figure supplement 3*). However, this antibody was less potent than POM1 and D18, since the currents were much smaller, and higher concentrations were required to produce consistent effects (66.7 nM for 6D11, compared to 33.3 nM for POM1 and 16.7 nM for D18) (*Figure 4—figure supplement 3* and *Figure 4*). 6D11 and three other antibodies targeting this region, D13 (*Williamson et al., 1998*) (epitope: a.a. 95–105), POM3 (*Polymenidou et al., 2008*) (epitope: a.a. 95–100) and ICSM-35 (*Khalili-Shirazi et al., 2007*) (epitope: a.a. 93–105), did not induce currents at a concentration of 33.3 nM (*Figure 4—figure supplement 3*). Taken together, these results indicate that antibodies targeting a positively charged segment at the terminus of the flexible

domain are less effective at inducing currents than antibodies binding to helix 1 in the C-terminal, structured domain.

To explore further the epitope specificity of the antibody-induced currents, we tested two additional POM antibodies recognizing other epitopes in the C-terminal domain of PrP$^C$. We found that POM4 (whose epitope encompasses helix 3 and $\beta$1) and POM6 (which recognizes helices 1 and 2) did not produce detectable currents when applied to WT PrP-expressing N2a cells at a concentration of 33.3 nM (*Figure 4—figure supplement 4*).

We tested whether anti-PrP antibodies induce currents in hippocampal neurons, as well as in N2a cells. Treatment with D18 (33.3 nM) induced large, spontaneous inward in hippocampal neurons cultured from Tga20 mice over-expressing wild-type PrP, and also increased the fragility of these neurons (*Figure 4—figure supplement 5*). Eight of 10 Tga20 neurons analyzed in the presence of D18 were lost to observation shortly after breaking the patch, usually after recording an initial inward current that did not return to baseline. This phenomenon is similar to what we previously observed in cultured neurons expressing $\Delta$CR PrP (*Biasini et al., 2013*), and may reflect a detrimental effect of the induced currents on the integrity of the neuronal membrane. In contrast, recordings from Tga20 neurons in the presence of D18 and PPS remained stable for 5 min without current activity (*Figure 4—figure supplement 5*). Seven out of 15 wild-type neurons analyzed in the presence of D18 exhibited spontaneous currents, with no lost cells. D18 did not induce any currents in neurons from $Prnp^{-/-}$ mice, which lack PrP expression.

## Antibodies recognizing the C-terminal domain of PrP$^C$ induce dendritic degeneration in hippocampal neurons

Given the correlation between $\Delta$CR PrP-induced currents and neurotoxicity, we hypothesized that anti-PrP antibodies might have neurotoxic effects on cultured hippocampal neurons as a result of the currents induced by these antibodies. To test this idea, we treated hippocampal neurons cultured from Tga20 mice with D18 (16.7 nM) or POM1 (33.3 nM) for 48 hr, and then stained them with an antibody to MAP2 to visualize changes in dendritic morphology. We observed that treatment with D18 or POM1 caused dendrites to assume a characteristic 'beaded' appearance, which is typical of several kinds of toxic insults, including hypoxia and glutamate-induced excitotoxicity (*Hasbani et al., 2001*) (*Figure 5A*). This effect was less in wild-type neurons and completely absent in $Prnp^{-/-}$ hippocampal neurons, indicating a dependence on the expression level of PrP$^C$ (*Figure 5A*). Treatment of neurons with non-specific mouse IgG had no effect on dendritic morphology (*Figure 5A*). Antibody 6D11, which weakly induced currents in WT PrP-over-expressing N2a cells at 66.7 nM (*Figure 4—figure supplement 3*) induced mild dendrite degeneration in Tga20 neurons (*Figure 5—figure supplement 1A*), while ICSM-35 at 33.3 nM, which did not induce currents, did not cause dendritic degeneration (*Figure 5—figure supplement 1B*). Similarly, POM4 (33.3 nM) and POM6 (33.3 nM) had no effect on dendritic morphology (*Figure 5—figure supplement 2*).

We tested whether antibody-induced dendritic changes, like currents, were dependent on the N-terminal domain of PrP$^C$. Supporting such a correlation, D18 had no effect on dendritic morphology of hippocampal neurons cultured from mice expressing $\Delta$23–31 or $\Delta$23–111 PrP (*Figure 6*), demonstrating that the N-terminal domain is essential for both the dendrotoxic and current-inducing effects of the antibody. In addition, we found that co-treatment of neurons with the N-terminal ligands PPS, 100B3, or POM11 abolished D18-induced dendritic degeneration (*Figure 5B–D*), analogous to the way these ligands inhibit D18-induced currents (*Figure 4A–C*).

Given the proposed use of a humanized version of ICSM-18 as an immunotherapeutic for prion and Alzheimer's diseases in patients (*Klyubin et al., 2014*), we tested whether ICSM-18 induces dendritic toxicity. We observed that treatment of neurons cultured from Tga20 mice with ICSM-18 (6.67 nM) for 48 hr caused significant beading of dendrites, similar to the effects of POM1 and D18 (*Figure 7A*). At higher concentrations (33.3 nM), ICSM-18 caused significant loss of neuronal cell bodies after 48 hr of treatment (not shown). In control experiments, no morphological effects were observed with non-specific IgG (6.67 nM), or after treatment of neurons from $Prnp^{-/-}$ mice with ICSM-18 (*Figure 7A*). ICSM-18 scFv (200 nM) also induced dendritic degeneration on neurons cultured from Tga20 mice but not $Prnp^{-/-}$ mice, although the effect was milder than for the ICSM-18 holo-antibody (*Figure 7B*).

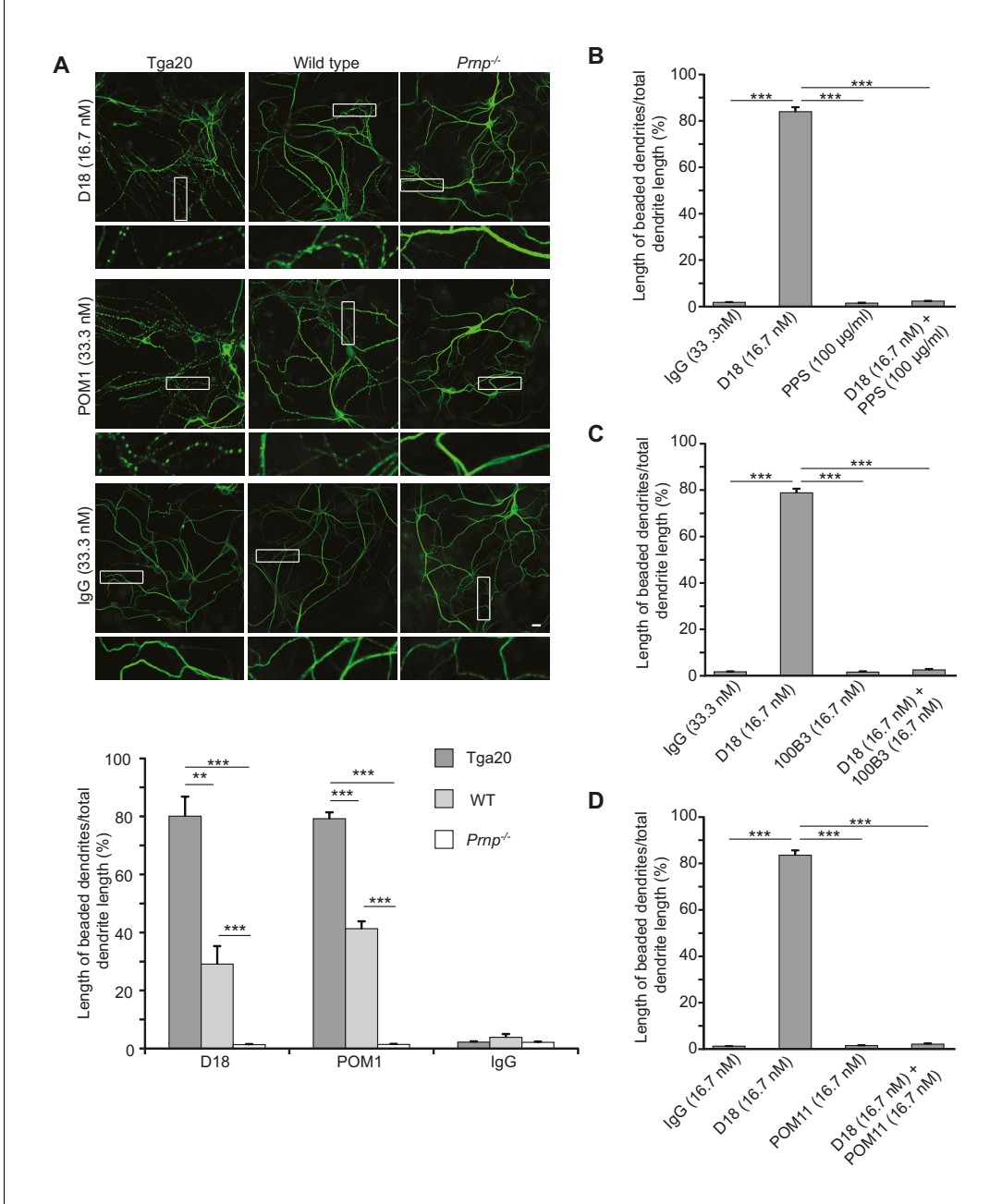

**Figure 5.** Antibodies recognizing the C-terminal domain of PrP[C] induce dendritic degeneration in hippocampal neurons. (**A**) Top, representative images showing dendrite morphology of cultured hippocampal neurons from Tga20 mice (which over-express WT PrP[C]), WT mice, or *Prnp*[−/−] mice after treatment for 48 hr with D18 (16.7 nM), POM1 (33.3 nM) or non-specific IgG (33.3 nM). The cells were stained with an antibody to MAP2 to visualize dendrites. Boxed areas are enlarged below each image. Scale bar = 10 μm. Bottom, quantitation of dendritic degeneration, expressed as the length of beaded dendrite segments as a percentage of total dendrite length, from 10 images in three independent cultures for each experimental condition. Data represent mean ± S.E.M. **p<0.01; ***p<0.005. (**B–D**) Quantitation of dendritic beading following treatment with IgG, D18 alone, N-terminal ligand (PPS, 100B3, or POM11) alone, or D18 together with the N-terminal ligand. Data represent mean ± S.E.M. ***p<0.005.

The following source data and figure supplements are available for figure 5:

**Source data 1.** Quantification of dendritic degeneration, expressed as the length of beaded dendrite segments as a percentage of total dendrite length.

**Figure supplement 1.** 6D11, but not ICSM35, has a weak effect on the dendritic morphology of Tga20 hippocampal neurons.

**Figure supplement 2.** POM4 and POM6 have no effect on the dendritic morphology of Tga20 hippocampal neurons.

*Figure 5 continued on next page*

*Figure 5 continued*

## Discussion

In this study, we have investigated two potentially interrelated functional activities of the PrP$^C$ molecule: its ability to induce ionic currents, and its ability to cause degenerative changes in neurons. Both activities are associated with deletions spanning the central region of the protein (residues 105–125), as well as with binding of antibodies to overlapping epitopes on the outer surface of helix 1 in the structured, C-terminal domain of the protein. Both activities are dependent on the flexible, N-terminal domain of PrP$^C$, and are blocked by deletion or mutation of this domain, or by binding of specific ligands to the N-terminal domain. When fused to an unrelated protein (EGFP), the N-terminal domain is sufficient by itself to induce current activity in the absence of the C-terminal domain. Taken together with evidence from heteronuclear NMR experiments, these results suggest a molecular model in which the N-terminal domain represents a toxic effector whose activity is regulated by

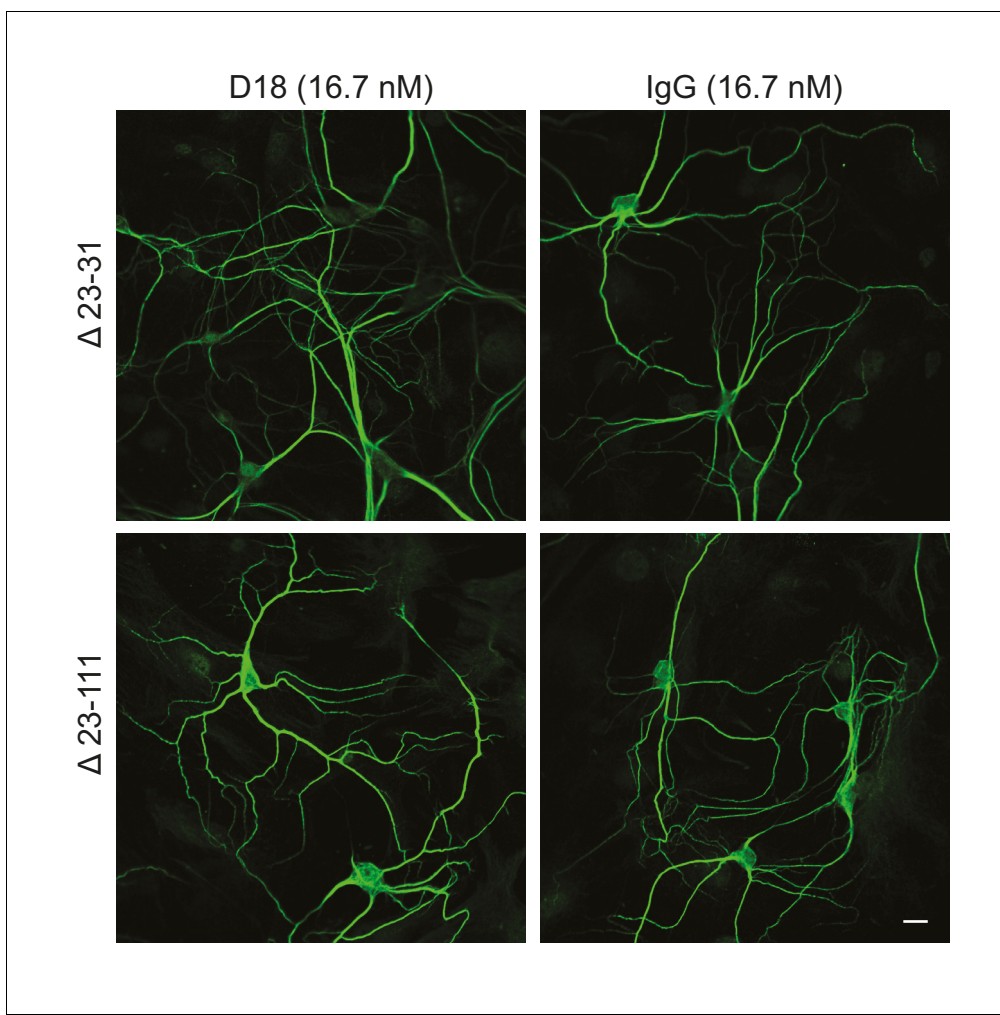

**Figure 6.** D18 does not induce dendritic degeneration in hippocampal neurons cultured from *Δ23–31* or *Δ23–111* transgenic mice on the *Prnp*$^{-/-}$ background. Representative images showing dendrite morphology of cultured hippocampal neurons from mice expressing *Δ23–31* PrP (upper panels) or *Δ23–111* PrP (lower panels) after treatment for 48 hr with D18 (16.7 nM) (left-hand panels) or non-specific IgG (16.7 nM) (right-hand panels). Neurons were stained for an antibody to MAP2 to visualize dendrites. Scale bar: 20 μm.

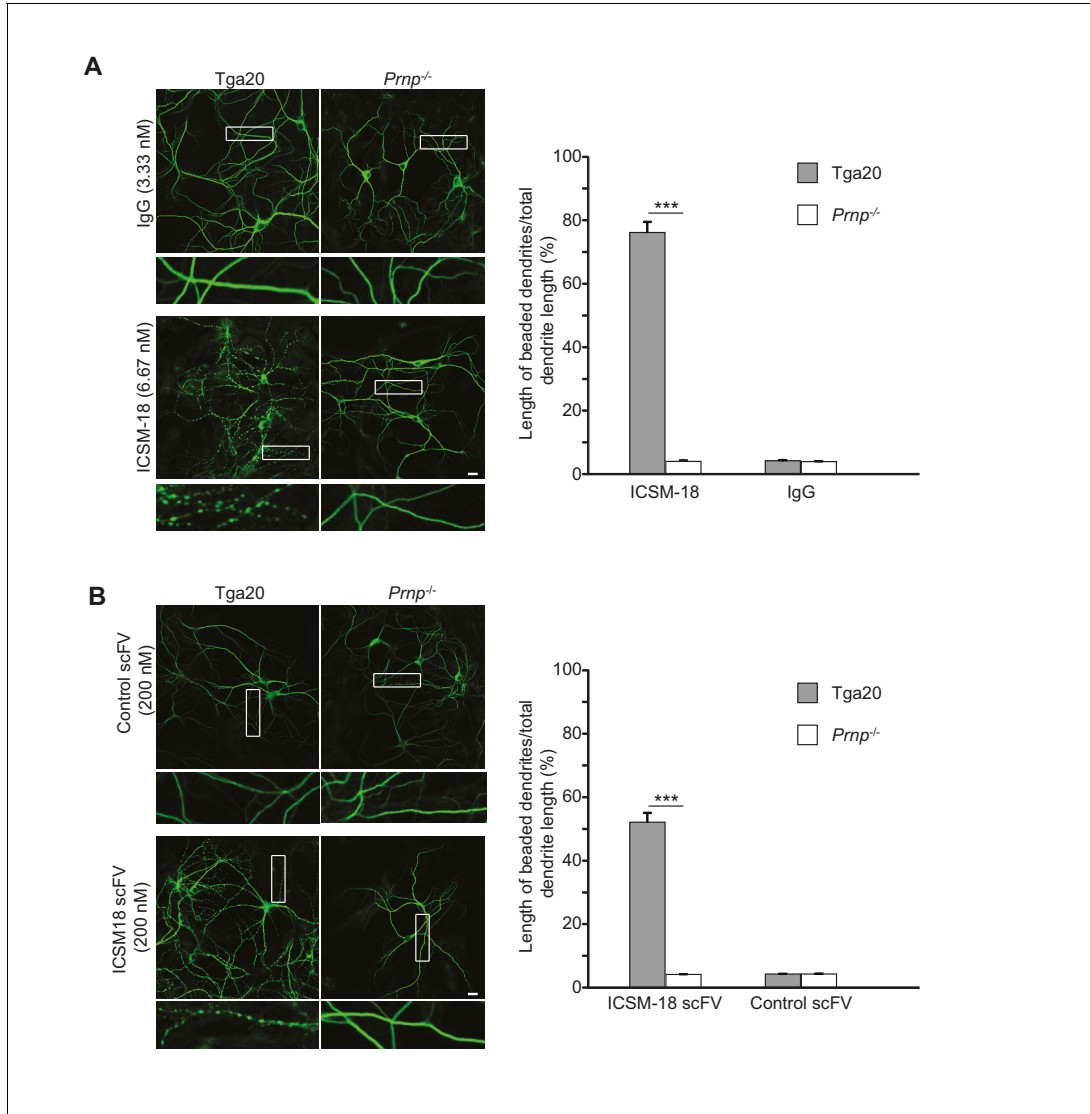

**Figure 7.** Anti-prion antibody ICSM-18 induces dendritic degeneration in hippocampal neurons. (A) Left, representative images showing dendrite morphology of cultured hippocampal neurons from Tga20 mice (which over-express WT PrP$^C$) (left-hand panels) or $Prnp^{-/-}$ mice (right-hand panels) after treatment for 48 hr with non-specific IgG (6.67 nM) (upper panels) or ICSM-18 (6.67 nM) (lower panels). The cells were stained with an antibody to MAP2 to visualize dendrites. Boxed areas are enlarged below each image. Scale bar = 10 µm. Right, quantitation of dendritic degeneration, expressed as the length of beaded dendrite segments as a percentage of total dendrite length, from 10 images in three independent cultures for each experimental condition. Data represent mean ± S.E.M. ***p<0.005. (B) Left, representative images showing dendrite morphology of cultured hippocampal neurons from Tga20 mice (left-hand panels) or $Prnp^{-/-}$ mice (right-hand panels) after treatment for 48 hr with control scFv (anti-fluorescein, 200 nM) (upper panels) or ICSM-18 scFv (200 nM) (lower panels). The cells were stained with an antibody to MAP2 to visualize dendrites. Boxed areas are enlarged below each image. Scale bar = 10 µm. Right, quantitation of dendritic degeneration, expressed as the length of beaded dendrite segments as a percentage of total dendrite length, from 10 images in three independent cultures for each experimental condition. Data represent mean ± S.E.M. ***p<0.005.

the C-terminal domain, most likely by a Cu$^{2+}$-promoted physical interaction between the two domains. We speculate that alterations of this intramolecular regulation may have both pathological and physiological consequences.

## The N-terminal domain of PrP$^C$ is necessary and sufficient for current activity

Our previous studies identified a nine amino-acid polybasic region at the N-terminus of PrP$^C$ (residues 23–31) that is essential for several toxic activities, including the spontaneous current activity associated with the $\Delta$CR PrP deletion mutant, the antibiotic hypersensitivity of cells expressing $\Delta$CR PrP, and the neurodegenerative phenotype of transgenic mice expressing the deletion mutant $\Delta$34–121 (*Solomon et al., 2011*; *Westergard et al., 2011b*). In the present study, we have shown that reversal of positive charges within this region (three lysine residues and one arginine residue) abolishes $\Delta$CR current activity, as does treatment with ligands (antibodies, Cu$^{2+}$ ions, pentosan sulfate) that bind to this region or to other sites within the flexible N-terminal domain (residues 23–125). Strikingly, the isolated N-terminal domain fused to an unrelated protein (EGFP) has the ability to induce spontaneous ionic currents. The magnitude of these currents is quantitatively related to the length of the N-terminus incorporated (fusions ending at residues 31, 58, 90, and 109 produce progressively more current), and the activity of these constructs is entirely dependent on the presence of the 23–31 region. Taken together, these results suggest that the N-terminal domain of PrP$^C$ acts as an autonomous effector of ionic current activity, and that basic amino acids at the extreme N-terminus are essential for this activity.

We suggest two possible models to explain how the N-terminal domain induces currents. One model is based on the fact that polybasic residues 23–31 resemble a 'protein transduction domain', originally described in the HIV Tat protein (*Wadia et al., 2008*). Such positively charged domains are capable of penetrating lipid bilayers and creating pores, by virtue of binding to and disrupting membrane phospholipids (*Herce and Garcia, 2007*). The N-terminal domain of PrP$^C$ may thus function as a 'tethered' protein transduction domain capable of spontaneously and transiently penetrating the lipid bilayer to produce pores that allow passage of ions. Consistent with penetration of a positively charged protein domain into or across the cell membrane, we find that the ionic current activity associated with both $\Delta$CR and PrP(N)-EGFP-GPI is apparent only at hyperpolarized holding potentials ($<-30$ mV), similar to the resting potential of neurons ($-60$ to $-70$ mV). A second possibility is that the N-terminal domain interacts with other membrane proteins, for example endogenous ion channels or channel-modulating proteins, to induce current activity. Supporting this hypothesis is our observation that co-expression of wild-type PrP$^C$ suppresses the ionic currents induced by $\Delta$CR and PrP(N)-EGFP-GPI, a phenomenon that could be attributable to competition between the wild-type and mutant forms for a common membrane-associated target protein.

## Antibodies to the C-terminal domain induce currents

Three different antibodies (POM1, D18, ICSM-18) targeting the C-terminal domain all induce ionic current activity in cells expressing WT PrP$^C$. Importantly, the properties of these currents are identical to those of the spontaneous currents associated with $\Delta$CR PrP, in terms of their sporadic nature, their blockage by N-terminal ligands (PPS, antibodies), and their absolute dependence on the presence of the polybasic 23–31 region. Based on crystal structures, mutagenesis studies, and peptide arrays, the epitopes of these antibodies, while not identical, are largely overlapping and encompass the outer surface of helix 1, as well as parts of the $\beta$1-$\alpha$1 loop and helix 3 in the case of POM1 (*Antonyuk et al., 2009*; *Doolan and Colby, 2015*; *Sonati et al., 2013*). Three different antibodies (POM3, D13 and ICSM-35) recognizing a basic region following the octapeptide repeats (residues 93–110) were ineffective at inducing currents, although a fourth one (6D11) had a weak effect at high concentrations. Two other antibodies targeting additional epitopes in the C-terminal domain had no effect. *Table 1* summarizes the effects of the antibodies on current activity.

Together, these findings suggest a novel intramolecular regulatory mechanism controlling the activity of PrP$^C$ (*Figure 8A*). Published crystal structures of both POM1 (*Sonati et al., 2013*) and ICSM-18 (*Antonyuk et al., 2009*) bound to PrP$^C$ indicate that these antibodies do not induce major structural alterations in the PrP$^C$ globular domain compared to the unliganded state, arguing against antibody-induced allosteric changes as a toxic mechanism. Rather, our results suggest that antibodies bound to helix 1 disrupt a critical regulatory interaction between the N- and C-terminal domains, thereby liberating the N-terminal domain to produce toxic effects (*Figure 8B*). The fact both Fab and scFv forms of the relevant antibodies display current-inducing activity suggests that antibody ligands as small as 25–50 kDa are able to disrupt N-C interactions. We propose that a similar loss of

**Table 1.** Anti-prion antibodies used in this study, and their ability to induce ionic currents and dendritic toxicity.

| Antibody | Epitope | Currents | Dendritic toxicity |
|---|---|---|---|
| D18 (holo and Fab) | 132–156 (α1) | Yes | Yes |
| POM1 | 138-147/204,8,12 (α1/α3) | Yes | Yes |
| ICSM18 (holo and scFv) | 143–156 (α1) | Yes | Yes |
| 6D11 | 93-109/97-100 | Weak | Weak |
| ICSM35 | 93–105 | No | No |
| D13 | 95–105 | No | No |
| POM4 | 121-134/218-21 (β1/α3) | No | No |
| POM6 | 140/145/174/177 (α1/ α2) | No | No |
| POM11* | 51–90 (octarepeats) | No | No |

*Blocks currents induced by D18 and POM1.

regulation occurs when residues within the 95–125 region are deleted (as in ΔCR), or when an unrelated protein is substituted for the C-terminal domain (PrP(N)-EGFP-GPI) (*Figure 8C,D*). In this scenario, the toxic activity of the liberated N-terminal domain would be blocked by binding of ligands, including PPS, antibodies, or $Cu^{2+}$ ions (*Figure 8E*).

The model proposed in *Figure 8* is supported by biophysical evidence. NMR studies performed here as well as previously (*Evans et al., 2016*; *Spevacek et al., 2013*) demonstrate that the N-terminal domain docks onto the C-terminal domain, thereby regulating in cis the ability of the N-terminal domain to promote toxic effector functions under normal conditions. Of note, the docking site encompasses the POM1 epitope (*Evans et al., 2016*). Thus, binding of antibodies to this region would be predicted to disrupt the N-C interaction, leading to toxic activity. We also show here that deletion of the central region (in ΔCR PrP) weakens $Cu^{2+}$-induced N-C interaction, consistent with the hypothesis that the toxicity of the ΔCR mutant results from disrupted regulation of the N-terminal domain. $Cu^{2+}$ ions are physiological ligands of $PrP^{C}$ (*Millhauser, 2007*), and changes in endogenous $Cu^{2+}$ concentration are likely to modulate the strength of N-C interactions. $Cu^{2+}$ binding to the octarepeats may also directly suppress the toxic activity of the N-terminal domain, as we have observed for the currents induced by ΔCR PrP and PrP(N)-EGFP-GPI.

## C-terminally directed antibodies induce dendritic degeneration

In addition to acutely inducing ionic currents, helix 1 antibodies (POM1, D18, ICSM18) cause major changes in dendritic morphology of cultured hippocampal neurons, in particular the appearance of blebs or varicosities, when applied for longer periods of time (48 hr) (*Table 1*). These antibody-induced dendritic changes are, like the currents induced by the same antibodies, entirely dependent on the N-terminal domain of $PrP^{C}$, and are blocked by N-terminal ligands and deletions of residues 23–31.

The parallel characteristics of the ionic currents and the dendritic changes induced by C-terminal antibodies suggest a mechanistic connection between the two phenomena. One possibility is that chronic activation of $PrP^{C}$-mediated currents leads directly or indirectly to dendritic degeneration. Consistent with this possibility, dendritic varicosities similar to those caused by anti-PrP antibodies are a characteristic feature of glutamate excitotoxicity (*Hasbani et al., 2001*), and glutamate excitotoxicity has been implicated in the neuronal degeneration induced by ΔCR PrP (*Biasini et al., 2013*; *Christensen et al., 2010*). A second possibility is that current induction and dendritic degeneration are parallel events that represent two distinct outputs of antibody binding to $PrP^{C}$. For example, the N-terminal domain of $PrP^{C}$ may stimulate currents by interacting with the lipid bilayer, and dendritic changes by interacting with signal-transducing proteins embedded in the membrane.

Several other studies have reported toxic effects of anti-PrP antibodies, including several of the ones used here, although some of these results have been contradictory. It has been reported that antibodies D13 (*Reimann et al., 2016*; *Solforosi et al., 2004*), POM1 (*Sonati et al., 2013*), and

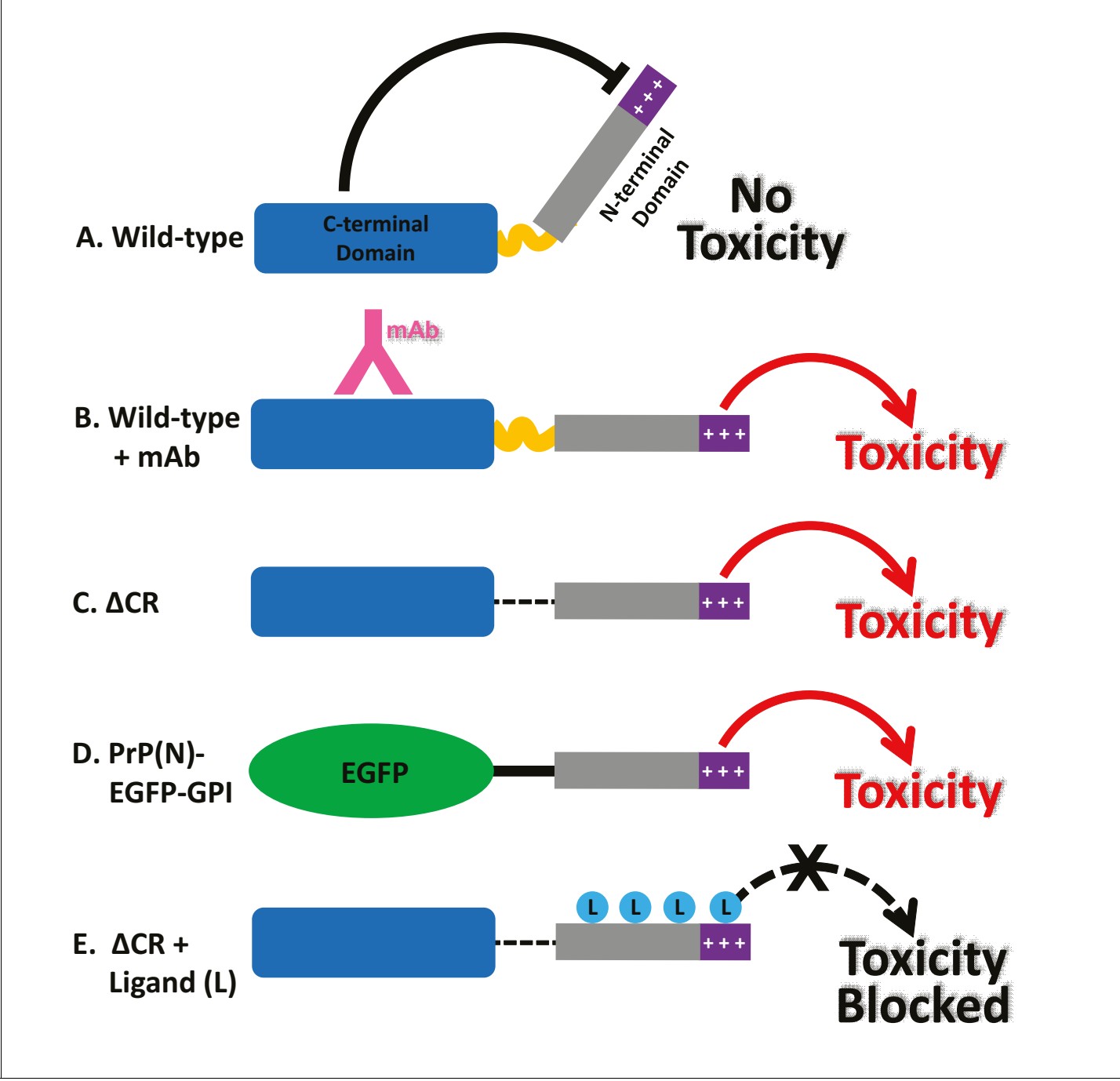

**Figure 8.** Models for the neurotoxic effects of PrP. (**A**) The C-terminal domain of PrP[C] negatively regulates the toxic effector function of the N-terminal domain. +++, basic residues within the 23–31 region at the extreme N-terminus, which are essential for the toxic action of PrP. (**B**) Binding of monoclonal antibodies to the C-terminal domain disrupts this regulatory interaction, releasing the N-terminal domain to produce toxic effects. (**C**) Deletion of the central region, as in ΔCR PrP, produces a similar loss of regulation, with toxic consequences. (**D**) When EGFP is substituted for the C-terminal domain of PrP[C], regulation is also lost. (**E**) Binding of ligands (PPS, antibodies, $Cu^{2+}$) to the N-terminal domain of ΔCR PrP blocks its ability to exert toxic effects. These ligands would have a similar, inhibitory effect on PrP(N)-EGFP-GPI (not shown).

ICSM-18 (*Reimann et al., 2016*), but not D18 (*Solforosi et al., 2004*), caused acute death (within 24–48 hr) of neurons when injected stereotaxically into the hippocampus or cerebellum. Using a similar protocol, however, *Klöhn et al. (2012)* found that D13 and ICSM-18 were non-toxic, as was ICSM-35. Finally, *Sonati et al. (2013)* observed that chronic treatment (10–21 days) of cerebellar slices with several C-terminally directed antibodies, including POM1, caused neuronal death, and they concluded that this effect was dependent on the flexible N-terminal domain. The results presented here are consistent with this proposal.

It has been reported that anti-PrP antibodies trigger several toxic mechanisms in cerebellar slices, including generation of reactive oxygen species, calpain activation and stimulation of the PERK arm of the unfolded protein response (*Sonati et al., 2013*). Whether these pathways are operative in our system remains to be determined. Since we observe relatively acute changes in dendritic morphology without loss of neuronal viability, it is possible that the downstream toxic pathways engaged by antibody treatment in our system may be different.

### Implications

The results presented here have several important implications. First, they add to concerns that have been raised regarding the use of anti-PrP antibodies as therapeutic tools for treatment of prion diseases (*White et al., 2003*) and Alzheimer's disease (*Klyubin et al., 2014*), given the potential side-effects of these reagents on neuronal viability at nanomolar concentrations (*Reimann et al., 2016*; *Solforosi et al., 2004*; *Sonati et al., 2013*).

Second, our results also suggest a mechanism by which pathologic ligands that bind to $PrP^C$ could produce neurotoxic effects by disrupting the normal regulatory cis-interaction between the N- and C-terminal domains, similar to the action of the antibodies described in this study. The Alzheimer's A$\beta$ peptide, which binds to $PrP^C$ and triggers functional and structural alterations in synaptic transmission (*Laurén et al., 2009*), is an example of such a ligand. $PrP^{Sc}$, which also binds to $PrP^C$ (*Solforosi et al., 2007*), might act in a similar fashion, although the fact that mice expressing N-terminally truncated $PrP^C$ remain susceptible to prion diseases (*Supattapone et al., 2001*; *Turnbaugh et al., 2012*) argues against this as the primary pathogenic mechanism in these disorders.

Finally, it is possible that the antibody-induced effects we have observed here are a reflection of a physiological activity of $PrP^C$. If so, natural ligands, including proteins, small molecules or metal ions, may exist, whose binding to $PrP^C$ regulates an effector activity of the N-terminal domain, similar to the way that we suppose anti-PrP antibodies operate. Copper ions are examples of natural ligands, binding of which to the octapeptide repeats promotes docking of the N- and C-terminal domains (*Evans et al., 2016*; *Spevacek et al., 2013*). Endogenous ligands for the globular domain may also exist, which either enhance or disrupt N-C interaction. Previous studies have implicated the N-terminal domain of $PrP^C$ in several physiological activities of $PrP^C$ (*Parkin et al., 2007*; *Pauly and Harris, 1998*; *Sempou et al., 2016*; *Shyng et al., 1995b*; *Taylor et al., 2005*; *Watt et al., 2012*), some of which may be regulated by interaction with the C-terminal domain.

## Materials and methods

### Antibodies and other reagents

POM1, POM3, POM4, POM6 and POM11 antibodies (*Polymenidou et al., 2008*) were provided to J.T. by the University of Zürich, Institute of Neuropathology. Hybridomas producing the human-mouse chimeric monoclonal antibodies D13 and D18 (*Safar et al., 2002*; *Williamson et al., 1998*) were provided by Anthony Williamson, Dennis Burton, and Bruce Chesebro. Antibodies were affinity-purified using protein A spin columns (Montage Antibody Purification Kit, EMD Millipore). Fab fragments of D18 were prepared using the Pierce Fab preparation kit from Thermo Scientific. The purity of all antibody preparations was verified by SDS-PAGE. The following antibodies were purchased from commercial sources: 100B3 (Wagening UR, Netherlands); 6D11 (BioLegend, cat #808001); ICSM-18 and ICSM-35 (D-Gen Ltd.).

Pentosan polysulfate (average MW = 4500–5000) was purchased from Biopharm Australia Pty Ltd., and phosphatidylinositol-specific phospholipase C from Sigma (cat #P5542).

## Expression and purification scFv antibodies

The genes encoding the scFv form of ICSM18, constructed as described (*Doolan and Colby, 2015*), and anti-fluorescein 4-4-20 scFv (negative control; obtained from Anne S. Robinson) were subcloned into pHAGE-CMV-dsRed-UBC-GFP-W (Addgene plasmid #24526) in place of dsRed by restriction digest cloning such that the final construct included an N-terminal Igκ light chain secretion signal and a C-terminal FLAG tag (DYKDDDDK) fused to the scFv, while maintaining the separate ORF expressing GFP as a reporter. Lentivirus was generated by co-transfection of $5 \times 10^5$ HEK cells with 1.3 µg pHAGE expression vector, 1 µg psPax2, and 0.65 µg pMD2.G (Addgene plasmids #12260 and #12259) with TransIT-293 transfection regent (Mirus Bio LLC). After 72 hr, the supernatant was filtered with a 0.45-µm syringe-tip filter and used directly for transduction.

CHO-S cells adapted to serum-free suspension culture (kindly provided by Kelvin H. Lee) were grown in serum-free culture medium (Thermo Fisher Scientific, SH3054902) in vent-cap shake flasks in a humidified incubator at 37°C/5% CO2. Stable CHO cell lines were created by transducing $5 \times 10^5$ CHO cells in 9 ml with 1 ml of HEK supernatant containing lentiviral particles in the presence 6 µg/ml polybrene (Sigma-Aldrich) for 2 days. Transduced cells were separated from non-transduced cells based on GFP reporter expression by fluorescence-activated cell sorting (FACS) using a BD FACSAria II FACS machine. Production of scFv was carried out by culturing the CHO lines for 7 days; initial cultures contained $5 \times 10^5$ cells/ml. Supernatant was harvested by centrifugation and filtration with a 0.22-µm syringe-tip filter before purification by anti-FLAG-affinity chromatography (Sigma-Aldrich) and low-pH elution according to the manufacturer's protocol. Eluted scFv was concentrated and buffer-exchanged with a 10 kDa molecular weight cut-off centrifugal filter, assayed by silver stain to ensure purity >90%, and quantified by Bradford assay. Purified scFv was stored in Tris-buffered saline at 4°C for short-term use, or −20°C for long-term storage.

## Plasmids

pcDNA3.1(+)Hygro plasmids (Invitrogen) encoding WT, ΔCR, ΔCR/Δ51–90, or Δ23–31 PrP have been described previously (*Solomon et al., 2010*, *2011*; *Turnbaugh et al., 2011*). PrP(N)-EGFP-GPI constructs were created by fusing DNA sequences encoding residues 1–31, 1–58, 1–90, 1–109 or 32–109 of mouse PrP to residues 1–239 of EGFP, followed by a poly-glycine/serine linker (GGGGS)₄ and then by residues 222–254 of mouse PrP in order to maintain GPI anchoring of the fusion protein. The ΔCR/E3D plasmid encodes ΔCR PrP with the following mutations: K23E, K24E, R25D and K27E.

## Cells

N2a cells (Cat. #: ATCC CCL-131, RRID: CVCL_0470) were maintained in DMEM supplemented with nonessential amino acids, 10% fetal bovine serum and penicillin/streptomycin. The N2a cell line we used in this study is mycoplasma free. Cells were transiently transfected using Lipofectamine 2000 with pEGFP-N1 (Clontech), along with empty pcDNA3.1(+)Hygro vector, or vector encoding WT or mutant PrPs. Cell-surface expression of all PrP constructs was confirmed by immunofluorescence staining.

Hippocampi from the newborn pups of the indicated genotypes were dissected and treated with 0.25% trypsin at 37°C for 12 min (*Shen et al., 2006*). Cells were plated at a density of 65,000 cells/cm² on poly-D-lysine-coated coverslips in DMEM medium with 10% F12 and 10% FBS.

## Mice

*Prnp*⁻/⁻ (Zurich I) mice (*Büeler et al., 1992*) on the C57BL6 background, and Tga20 mice (*Fischer et al., 1996*) have been described previously, and were obtained from EMMA (European Mouse Mutant Archive). Tg(PrPΔ23–31) mice (*Turnbaugh et al., 2011*) and Tg(PrPΔ23–111) mice (*Westergard et al., 2011a*) have been described previously, and were maintained on a *Prnp*⁻/⁻ background. Wild-type C57BL6 mice were obtained from Charles River Laboratories.

## Electrophysiological analysis

Recordings were made from N2a cells 24–48 hr after transfection. Transfected cells were recognized by green fluorescence resulting from co-transfection with pEGFP-N1. Hippocampal neurons were analyzed after 13–15 days in culture. Whole-cell patch clamp recordings were collected using standard techniques. Pipettes were pulled from borosilicate glass and polished to an open resistance of

2–5 megaohms. Experiments were conducted at room temperature with the following solutions: internal, 140 mM Cs-glucuronate, 5 mM CsCl, 4 mM MgATP, 1 mM Na$_2$GTP, 10 mM EGTA, and 10 mM HEPES (pH 7.4 with CsOH); external, 150 mM NaCl, 4 mM KCl, 2 mM CaCl$_2$, 2 mM MgCl$_2$, 10 mM glucose, and 10 mM HEPES (pH 7.4 with NaOH). Current signals were collected from a Multiclamp 700B amplifier (Molecular Devices, Sunnyvale, CA), digitized with a Digidata 1440 interface (Molecular Devices), and saved to disc for analysis with PClamp 10 software.

## Localization of PrP$^C$ on N2a cells

Immunofluorescence staining of cell surface PrP$^C$ on N2a cells was performed by incubating livings cells on ice with D18 antibody, fixing in 4% paraformaldehyde in PBS, and then labeling with Alexa Fluor 594 goat anti-human IgG (Molecular Probes, Eugene, OR). Images of N2a cells were acquired with a Zeiss LSM 710 confocal microscope.

## Morphological analysis of hippocampal neurons

Neurons cultured for 14 days were fixed with 4% paraformaldehyde in PBS and permeabilized with 0.2% Triton100. Fixed cultures were then incubated with primary antibodies against microtubule-activated protein 2 (MAP2; polyclonal; 1:1000; Abcam), followed by fluorescent secondary antibody. Confocal microscopic analysis was performed on a Zeiss LSM 710 microscope using a 20X objective lens. Identical acquisition settings were applied to all samples of the experiment. Images were analyzed with the NIH Image J program.

## Preparation of recombinant PrP for NMR

Recombinant PrP constructs encoding wild-type mouse PrP(23-230) and mouse ΔCR PrP(Δ105–125) in the pJ414 vector (DNA 2.0) were expressed in *E. coli* (BL21 (DE3) Invitrogen) (*Evans et al., 2016*). Mutations were introduced using PCR-based, site-directed mutagenesis with mutagenic primers (Invitrogen) and Phusion DNA Polymerase (Finnzymes). All constructs were confirmed by DNA sequencing.

Bacteria were grown in M9 minimal media supplemented with $^{15}$NH$_4$Cl (1 g/L) for $^1$H-$^{15}$N HSQC experiments or $^{15}$NH$_4$Cl and $^{13}$C6-glucose (2.5 g/L) for $^1$H, $^{13}$C, $^{15}$N triple-resonance experiments (Cambridge Isotopes). Cells were grown at 37°C until reaching an optical density (OD) of 0.6, at which point expression was induced with 1 mM isopropyl-1-thio-D-galactopyranoside (IPTG). PrP constructs were purified as previously described (*Spevacek et al., 2013*). Briefly, proteins were extracted from inclusion bodies with 8 M guanidium chloride (GdnHCl) (pH 8) at room temperature and were purified by Ni$^{2+}$-immobilized metal-ion chromatography (IMAC). Proteins were eluted from the IMAC column in 5 M GdnHCl (pH 4.5) and were brought to pH 8 with KOH and left at 4°C for 2 days to oxidize the native disulfide bond. Proteins were then desalted into 10 mM KOAc buffer (pH 4.5) and purified by reverse-phase HPLC on a C4 column. The purity and identity of all constructs were verified by analytical HPLC and mass spectrometry (ESI-MS). Disulfide oxidation was confirmed by reaction with N-ethylmaleimide and subsequent ESI-MS analysis.

Lyophilized protein samples were dissolved in degassed Milli-Q-purified H$_2$O and allowed to fully solubilize prior to all experiments. Protein concentrations were determined from the absorbance at 280 nm in GdnHCl buffer using an extinction coefficient of 64,840 M$^{-1}$ cm$^{-1}$ calculated for mouse PrP(23-230) and 62,280 M$^{-1}$cm$^{-1}$ calculated for mouse ΔCR PrP(Δ105–125). For samples containing Cu$^{2+}$, the metal ion was added from stock solutions of Cu(OAc)$_2$ or CuCl$_2$ in H$_2$O in which the Cu$^{2+}$ concentrations were accurately determined by EPR integration (*Walter et al., 2006*). Samples for NMR contained 200–400 μM PrP in 10 mM MES buffer (pH 6.1) with 10% D$_2$O.

## $^1$H-$^{15}$N HSQC NMR

NMR experiments were conducted on a Varian INOVA 600 MHz spectrometer equipped with a $^1$H, $^{13}$C, $^{15}$N triple-resonance cryoprobe. Resonance assignments were first obtained using standard triple-resonance experiments with a 400 μM samples of uniformly $^{13}$C,$^{15}$N-labeled mouse PrP(23-230) sample at 25°C. Experiments included HNCO, HN(CA)CO, HNCACB, CBCA(CO)NH, and CC(CO)NH. The assignments were then transferred to the $^1$H-$^{15}$N HSQC spectrum at 300 μM and 37°C by following the cross-peaks through concentration and temperature titrations. Resonances were confirmed, and additional resonances were assigned by recording three-dimensional HNCACB and $^{15}$N-

NOESY-HSQC spectra at 300 μM and 37°C. Finally, resonance assignments were transferred to the $^1H,^{15}N$ HSQC spectrum of MoPrP (H95Y/H110Y) at 300 μM and 37°C by visual inspection. All NMR spectra were processed with NMRPipe and NMRDraw (*Delaglio et al., 1995*), and analyzed using Sparky NMR Analysis and CcpNmr Analysis (*Vranken et al., 2005*).

$^1H,^{15}N$ HSQC spectra were recorded for MoPrP (H95Y/H110Y) and MoPrP(Δ105–125)H95Y constructs (300 μM) at 37°C both in the absence of metal ions and in the presence of 300 μM $CuCl_2$, and the HSQC peak intensities were determined using Sparky NMR Analysis and CcpNmr Analysis. Intensity ratios were analyzed using Kaleidagraph (Synergy Software) and residues with intensity reductions greater the one standard deviation of the mean were considered significantly perturbed.

## Acknowledgements

This work was supported by NIH grants R01 NS065244 (to DAH), R01 GM065790 (to GLM), and GM104316 (to DWC); NSF grant 1454508 (to DWC); and by funds from the German Research Foundation (TA 167/6) to JT We thank Dr. Gerold Schmitt-Ulms for supplying *Prnp* gene-edited N2a cells; Anthony Williamson, Dennis Burton, and Bruce Chesebro for providing D13 and D18 hybridoma cells; and the University of Zürich, Institute of Neuropathology, Switzerland, for providing the POM antibodies.

## Additional information

### Funding

| Funder | Grant reference number | Author |
|---|---|---|
| National Institutes of Health | R01 NS065244 | Bei Wu<br>Alex J McDonald<br>Celeste B Rich<br>David A Harris |
| National Institutes of Health | R01 GM065790 | Kathleen Markham<br>Glenn L Millhauser |
| National Institutes of Health | GM104316 | Kyle P McHugh<br>David W Colby |
| National Science Foundation | Grant 1454508 | Kyle P McHugh<br>David W Colby |
| Deutsche Forschungsgemeinschaft | (TA 167/6) | Jörg Tatzelt |

N.I.H. R01 NS065244 to D.A.H had a role in study design, data collection and interpretation. N.I.H. R01 GM065790 to G.L.M. had a role in data collection. N.I.H. GM104316 to D.W.C. and N.S.F. grant 1454508 to D.W.C. had a role in data collection. German Research Foundation (TA 167/6) to J.T. had a role in data collection.

### Author contributions

BW, Conceptualization, Data curation, Formal analysis, Investigation, Methodology, Writing—original draft; AJM, Conceptualization, Investigation, Writing—original draft; KM, CBR, Investigation; KPM, JT, DWC, Resources; GLM, Supervision; DAH, Supervision, Writing—original draft

### Author ORCIDs

Bei Wu, http://orcid.org/0000-0002-3368-2365
Alex J McDonald, http://orcid.org/0000-0003-1214-3562
Jörg Tatzelt, http://orcid.org/0000-0001-5017-5528
David A Harris, http://orcid.org/0000-0002-6985-5790

### Ethics

Animal experimentation: This study was performed in strict accordance with the recommendations in the Guide for the Care and Use of Laboratory Animals of the National Institutes of Health. All of the animals were handled according to approved institutional animal care and use committee (IACUC) protocols (#AN14997) of Boston University.

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
