## [Decision Letter]

Thank you for submitting your article "The N-terminus of the Prion Protein Is a Toxic Effector Regulated by the C-terminus" for consideration by *eLife*. Your article has been favorably evaluated by Rob Krumlauf (Senior Editor) and four reviewers, one of whom is a member of our Board of Reviewing Editors. The following individual involved in review of your submission has agreed to reveal his identity: Surachai Supattapone (Reviewer #3).

The reviewers have discussed the reviews with one another and the Reviewing Editor has drafted this decision to help you prepare a revised submission.

Summary:

In this manuscript, the authors present data from multiple complementary approaches, and build upon prior results, to elucidate the role of the N-terminal domain of PrP in mediating ionic membrane currents and neuronal degeneration. The reviewers are enthusiastic about insights into a potential mechanism for the toxicity of certain anti-PrP antibodies, which may help to resolve the current controversy about using these antibodies in human immunotherapy trials. The model may help explain the toxicity of Abeta oligomers, which bind to PrPC, although this isn't directly tested. The reviewers have elaborated several questions and concerns that must be addressed before this manuscript can be considered for publication.

Essential revisions:

1) The reviewers point out two apparent inconsistencies that need resolution: First, the failure of the 1-31-EGFP and 1-59-EGFP constructs to induce currents is puzzling since the deltaCR/delta51-90 mutant is very active in this regard and nearly identical to these EGFP constructs in terms of the N-terminal domain. Second, the lack of antibody-induced currents or toxicity in the delta23-31 cells and transgenic mouse neurons is surprising since one would expect that wild-type levels of PrP are present.

2) The relevance of the current findings to disease is incompletely explored. Whereas the connection between the toxicity of the POM1 and d13 antibodies with the ionic current mechanism is novel and interesting, the authors also speculate that the N-terminus of PrPC may be the mediator of PrPSc- and Abeta-induced neurodegeneration. This can be addressed by emphasizing that the mechanism explored relates to antibody-induced current only. Alternatively, to support the relationship to PrPSc- and Abeta-induced neurodegeneration, the reviewers have requested further elucidation of this connection and suggested additional experiments in this regard. Specifically, one reviewer expressed skepticism that the N-terminus of PrPC is required for the PrPSc-induced neurodegeneration since transgenic mice expressing N-terminally truncated PrPC develop spongiform degeneration following prion inoculation (J Virol. 2001 Feb;75(3):1408-13). On the other hand, perhaps the N-terminus of PrPC is required for Abeta-mediated toxicity, and we encourage the authors to test this directly (e.g. does application of Abeta oligomer cause the current?) Also, do recombinant PrP oligomers or fibrils induce currents?

*Reviewer #1:*

The PrPc protein harbors an intrinsically disordered segment between aa23-125. It was previously noted that expression in mice of a mutant form of PrPc with internal deletions of aa102-125 result in degeneration in the absence of PrP amyloid assembly and, furthermore, that this toxicity is suppressed in a dose-dependent manner by OE of wild type PrPc. In parallel it was noted that expression of internal deletion mutants of PrPc in cell lines induce spontaneous ionic currents that are also suppressed by OE of wild type PrPc, although a cause-and-effect relationship between ionic current induction and degeneration remains to be established. The current study focuses on the role of the N-terminus of PrPc in mediating the toxicity.

Within the N-terminus of PrPc, prior studies suggested that a charged region (aa23-31) is essential for current induction by expressing PrPc-delta105-125 in cells and also degeneration in mice expressing PrPc-delta32-134. Prior studies also showed that treatment of cells with pentosan polysulfate (PPS, which purportedly binds to the N-terminus of PrPc) also abolished current induction. Here the investigators show that treatment with anti-PrPC antibodies 100B3 (epitope aa24-28) and POM11 (epitope aa51-90) both reduce current induction in N2a cells expressing mutant deltaCR PrPc. Deletion of the epitope for POM11, a region of octapeptide repeats, eliminates the effect of POM11 treatment. Further investigating the octapeptide repeats they use show that addition of Cu++, which purportedly binds the octapeptide repeats reduces current induction in N2a cells expressing mutant deltaCR PrPc, similar to antibody binding. Further confirming the importance of the positively charged region, a mutant (E3D, in which 3 charges are reversed) also abolished current-inducing activity. Interestingly, it is shown that the N-terminal domain itself (aa1-109), in the absence of the C-terminal domain, is sufficient to induce spontaneous currents. This activity reduces with smaller fragments. These currents are inhibited by ligands, including PPS, Cu++, and antibodies 100B3 and POM11. NMR-based interrogation suggests that aa102-125 deletion, which correlates with current induction, reduces interaction between the C-terminus and N-terminus of PrPc. Antibodies targeting the C-terminal folded domain also induced spontaneous currents, and expression of PrPc-delta23-31 were resistant to current induction. One antibody (D18) failed to induce currents in cells pre-treated with PIPLC to cleave PrPc, or in cells with PrPc knockout.

The proposal that a disinhibited native function of PrPc mediates the toxicity of PrPsc (and perhaps A-β), and that the mechanism relates to disturbance of association between the C-terminal and N-terminal domains of PrPc is certainly intriguing. As an outsider to the prion field, I find this line of investigation fascinating. With respect to the current work, I find compelling the evidence that the C-terminus of PrPc suppresses an activity of the N-terminus that underlies the induction of a spontaneous ionic current. I have several experimental questions, but the two more important issues to me are:

1) what is the disease relevance of the spontaneous currents examined here? and 2) to what extent does this work provide a conceptual advance or mechanistic insight beyond what was previously known? First, with respect to the spontaneous currents elicited by expression of mutant PrPc (PrPc-delta105-125), as far as I can tell there is no established cause-and-effect relationship with degeneration, merely a correlation between induction of these currents in cell culture and degeneration in cells or in mice. Even if it is clear that spontaneous currents associated with forms of PrPc with internal deletions can drive toxicity, is this toxic gain of function relevant and essential to degeneration driven by PrPsc (and A-β)? Second, it appears that prior work has already established that PrPc deletion mutants induce spontaneous currents that are mitigated by N-terminal binding ligands, that the N-terminus interacts with the C-terminus, and that similar currents can be induced by antibodies targeting certain C-terminal epitopes. Just how much of a mechanistic advance does the current paper represent?

*Reviewer #2:*

This important study by Wu and colleagues focuses on understanding the mechanism of neurotoxicity induced by PrPC. These new results shed light on how PrP can trigger ionic currents and neuronal degeneration, and also provide hints into the physiologic function of PrPC. Residues 23-26 in the N-terminus of PrP were previously identified as key to inducing spontaneous ionic currents in cells with deletions in the central region of PrP (105-125). Here, in a carefully controlled series of experiments, additional ligands specific for the N-terminus abolished spontaneous ionic currents, as did mutating the suspected N-terminal lysine residues. The C-terminus of PrP was not required for the induction of ionic currents. The experiments are well-designed and logical, and for the most part, clearly described. Inclusion of relevant negative data would be informative (noted below). Overall this work is a highly significant contribution toward understanding prion protein pathophysiology, and may have implications for AD since amyloid-β oligomers may bind PrPC and lead to neuronal toxicity.

Suggestions for improving the manuscript clarity are noted below.

1) Whether ionic currents are induced during prion disease is unclear. Did the authors test recPrP oligomers or fibrils for induction of currents?

2) Three antibodies targeting C-terminal epitopes in PrP were found to induce spontaneous currents, but it is unclear how many antibodies were tested. Were antibodies to other C-terminal epitopes tested (i.e. helix 3)? These would provide additional support for helix 1 specificity.

3) Further description of the NMR methods and analysis in the Results is necessary for clarity, i.e., what residues in the C-terminus were impacted by copper binding to the OR? This seems to be the key point of these results, yet the interaction patch is not described beyond helices 2 and 3 in the C-terminus. From the figure, it appears that there is no longer an interaction with helix 3. Also a brief explanation of the NMR spectra would be helpful. Also the reference to the methods used from the Evans et al. study is unclear, was there a key part of the methods employed?

4) The currents in untransfected N2a were reported as smaller, but the figure shows fewer (% of time). How much smaller were the currents in the untransfected cells?

5) Subsection “Antibodies against the C-terminal domain and hinge region of PrP^C^ induce ionic currents”, last paragraph – Were anti-PrP antibodies tested in WT hippocampal neurons tested in addition to the tga20 neurons? It would be important to know whether overexpression was required to observe currents or the dendritic degeneration in neurons. Please clarify.

6) Adriano Aguzzi and colleagues have published several high profile papers on prion-induced toxicity, also describing the N-terminus as an effector regulated by the C-terminal domain. It would be important to discuss how the results here correspond to these findings from the field.

7) In previous reports, the PERK arm of the UPR was activated following exposure to anti-PrP C-terminal antibodies (Hermann et al., 2015 PLoS Pathogens). Did the authors test whether the PERK pathway was activated in the hippocampal neurons? Or whether calpain activation occurred (as observed in the Sonati et al. study, Nature 2013)?

8) The beaded appearance of the dendrites in Figure 5 is difficult to appreciate in these small images.

9) Typo in Figure 2 – Figure shows results from a deletion of 32-109, but legend states deletion of 23-109.

*Reviewer #3:*

I think this is an outstanding manuscript. The authors use a multidisciplinary approach (electrophysiology, NMR, antibodies, mutagenesis, dendritic spine analysis) with appropriate controls to make an extraordinarily compelling case that the N-terminus of PrP mediates toxicity through inducing an ionic current in cells. The methods, data, rigor, and writing are all top-notch.

The work is very significant because it provides a mechanism for the toxicity of certain anti-PrP antibodies, which may help to resolve the current controversy about using these antibodies in human immunotherapy trials. The model may help explain the toxicity of Abeta oligomers, which bind to PrPC, although this isn't directly tested.

*Reviewer #4:*

This manuscript examines the role of the N-terminal domain of the prion protein in mediating ionic membrane currents and neuronal degeneration. Previous result showed that the currents induced by the deltaCR PrP variant (missing residues 105-125) depend on a cluster of positive charges near the very N-terminus of the protein (residues 23-31), that a ligand to the N-terminus of the protein (PPS) eliminates the currents. They now show that two N-terminal epitope antibodies (100B3 and POM11) also reduce these currents, as does copper-2 binding, and as does mutation of 4 positive charges in the positive charge cluster to negative charges (E3D mutant).

A construct of the N-terminal domain fused to EGFP also elicits ionic currents, but N-terminally and C-terminally truncated versions induced less currents. Notably, here it is unclear why 1-31-EGFP and 1-58-EGFP were not active, since the deltaCR/delta51-90 mutant is very active and nearly identical to these EGFP constructs in terms of the N-terminal domain?

NMR then shows that the N-to-C-terminal contacts already demonstrated by Millhauser et al. in a previous publication are reduced for the deltaCR mutant. Notably, the data for the WT (panels A1 in Figure 3—figure supplement 1 and panels A1, B1 and C1 in Figure 3) appear essentially identical to those previously published, raising the question of whether their inclusion is justified here, and at a minimum a reference must be included in the figure legend.

Antibodies to the folded C-terminal domain of PrP (POM1, D18 and ICSM-18), previously shown to be toxic, also induced currents that were suppressed by PPS and the POM11 antibody in cells expression WT PrP, but less current was induced for cells expressing PrP missing residues 23-31. The effects are eliminated by cleaving PrP from the membrane, and are observed in WT cells, including hippocampal neurons, but not in PrP KO cells. D18 and POM1 also caused changes in dendritic morphology. Interestingly, in N2A cells expressing delat23-31 or neurons from transgenic mice expressing PrP missing residues 23-31 or 23-111, no effect (currents or toxicity) was observed. However, would the WT protein in these cells/TG mice not be expected to be susceptible to the antibodies and to cause currents/toxicity/degeneration?

The authors conclude that the N-terminal domain of PrP is responsible for the ionic currents they observe, and propose either a TAT peptide like mechanism or via protein-protein interactions. The propose that these currents, which are induced by antibodies to the C-terminal folded domain, may be responsible for the toxicity of such antibodies, as well as the toxicity of PrpSC, which is also reported to bind to the C-terminal domain of PrpC.

This is an interesting paper, which attempts to tie together a variety of previous observations from these groups and others, namely that antibodies to PrP can be toxic, that the flexible N-terminus of the protein mediates toxicity as well as ionic currents, and that intra-molecular interactions between the N- and C-terminus may regulate the activity/toxicity of the N-terminus. However, much of this model has already been presented in the various prior studies, including previous NMR work and previous work on the deltaCR construct and on the 23-31 deletion. The work is logical extension of the prior studies, but falls out largely as might be expected. That said, there are some troubling discrepancies, including the failure of the 1-31-EGFP and 1-59-EGFP constructs to induce currents, the lack of antibody-induced currents/ toxicity in the delta23-31 cells and transgenic mouse neurons, and the lack of any data supporting a specific mechanism linking the ionic currents to cell toxicity.

[Editors' note: further revisions were requested prior to acceptance, as described below.]

Thank you for resubmitting your work entitled "The N-terminus of the Prion Protein Is a Toxic Effector Regulated by the C-terminus" for further consideration at *eLife*. Your revised article has been favorably evaluated by Huda Zoghbi (Senior Editor), a Reviewing Editor, and three reviewers.

The manuscript has been improved but there are some remaining issues that need to be addressed before acceptance, as outlined below:

Please revise the text of the manuscript to provide possible explanations for unanticipated results, as described by Reviewer #1.

*Reviewer #1:*

The authors have posited explanations for two of the deficits noted in my review of the manuscript, namely 1) that residues 91-104 may contribute to current induction in deltaCR/delta51-90 and are absent in 1-31-EGFP and 1-58-EGFP, or possibly that EGFP does not position 1-31 or 1-58 NTD sequences at the proper distance from the membrane; and 2) that no antibody-induced currents are observed in N2A cells transfected with delta23-31 PrP because this construct may compete for antibody binding or suppress endogenous PrP expression. These are valid potential explanations (though of course they could actually be tested!), but they do not make their way in any shape or form into the revised text. The authors should explicitly discuss these issues in the manuscript. Otherwise, despite the public review forum, these points will likely escape the notice of most readers.

*Reviewer #2:*

The authors have satisfactorily addressed the concerns I had with the initial manuscript. The questions I had have now been clarified. This paper is an important contribution to understanding the mechanisms of PrP-mediated neurotoxicity.

*Reviewer #3:*

I am satisfied with the authors' response to my comments.

---

## [Author Response]

*Essential revisions:*

*1) The reviewers point out two apparent inconsistencies that need resolution: First, the failure of the 1-31-EGFP and 1-59-EGFP constructs to induce currents is puzzling since the deltaCR/delta51-90 mutant is very active in this regard and nearly identical to these EGFP constructs in terms of the N-terminal domain.*

There are at least two possible reasons why ΔCR/Δ51-90 PrP (Figure 1 and Figure 1—figure supplement 1) was more effective at inducing currents than 1-31-EGFP or 1-59-EGFP (Figure 2). First, the ΔCR/Δ51-90 PrP construct contains additional sequences that are not present in the 1-31-EGFP or 1-59-EGFP constructs. In particular, ΔCR/Δ51-90 PrP contains residues 91-104, which are absent in 1-31-EGFP and 1-59-EGFP. These additional residues may enhance production of spontaneous currents, consistent with the general observation that PrP-EGFP chimeras incorporating longer stretches of the PrP N-terminus produced more currents (Figure 2). A second possible explanation is that the EGFP portion of the chimeric constructs may position the PrP N-terminus at a different distance from the membrane, or in a different orientation, than the natural PrP^C^ C-terminus, and this may diminish the ability of the N-terminus to interact with the membrane to produce currents. It was necessary to substitute the C-terminal domain of PrP with EGFP in order to facilitate delivery of the proteins to the cell surface.

*Second, the lack of antibody-induced currents or toxicity in the delta23-31 cells and transgenic mouse neurons is surprising since one would expect that wild-type levels of PrP are present.*

In Figure 6, the Δ23-31 and Δ23-111 PrP transgenic mice used to derive the neurons were maintained on a PrP knockout background, so there is no wild-type PrP^C^ in those neurons. This has now been clarified in the legend. The Material and Methods section in the original manuscript incorrectly stated that these mice were maintained on a C57BL6 background; this has now been corrected.

In N2a cells expressing Δ23-31 PrP, one would expect a significant reduction in antibody-induced currents compared to cells over-expressing WT PrP (as was, in fact, observed in Figure 4), since the endogenous level of WT PrP in N2a cells is low, and we have shown that untransfected cells display reduced currents after antibody treatment (Figure 4—figure supplement 2). The fact that the Δ23-31-expressing cells do not display even low levels of current due to endogenous PrP^C^ may be due to some suppression of endogenous PrP expression, or perhaps competition by the deleted protein for binding of the antibodies.

*2) The relevance of the current findings to disease is incompletely explored. Whereas the connection between the toxicity of the POM1 and d13 antibodies with the ionic current mechanism is novel and interesting, the authors also speculate that the N-terminus of PrPC may be the mediator of PrPSc- and Abeta-induced neurodegeneration. This can be addressed by emphasizing that the mechanism explored relates to antibody-induced current only. Alternatively, to support the relationship to PrPSc- and Abeta-induced neurodegeneration, the reviewers have requested further elucidation of this connection and suggested additional experiments in this regard. Specifically, one reviewer expressed skepticism that the N-terminus of PrPC is required for the PrPSc-induced neurodegeneration since transgenic mice expressing N-terminally truncated PrPC develop spongiform degeneration following prion inoculation (J Virol. 2001 Feb;75(3):1408-13). On the other hand, perhaps the N-terminus of PrPC is required for Abeta-mediated toxicity, and we encourage the authors to test this directly (e.g. does application of Abeta oligomer cause the current?) Also, do recombinant PrP oligomers or fibrils induce currents?*

We acknowledge that the proposed role of the PrP^C^ N-terminus in PrP^Sc^ and Aβ toxicity is speculative, and is not directly addressed by any of the experiments reported in the paper. We are currently performing experiments to test this hypothesis, but we feel that the results are properly the subject of another paper. It would seem perverse to omit completely any reference to the possible implications of our results for prion pathogenesis, particularly since this connection has already been proposed in two publications from the Aguzzi laboratory (Sonati et al., 2013; Herrmann et al., 2015). However, in line with the suggestion of the reviewers, we have revised the text of the Abstract, Introduction, and Discussion sections to downplay the connection to PrP^Sc^ and Aβ toxicity, and to emphasize the relevance of our findings to the use of anti-PrP antibodies in a clinical setting.

With regard to the role of the N-terminus in PrP^Sc^ toxicity, the reviewers rightly point out that mice expressing N-terminally truncated PrP^C^ are susceptible to prion disease (although with a protracted incubation time), as shown in the 2001 J. Virology publication, as well as in our own work (Turnbaugh et al., J. Neurosci. 32:8817– 8830, 2012). We have now cited these papers in the Discussion section, in order to qualify our suggestion about the role of the N-terminus in PrP^Sc^ toxicity. However, we wish to point out that the results of these two studies do not definitively rule out a role for the N-terminal domain in PrP^Sc^ toxicity. Multiple pathogenic mechanisms likely contribute to neurodegeneration at different times during the course of chronic disorders like prion diseases, and eliminating one of these mechanisms (such as toxicity mediated by the PrP^C^ N-terminus) would still allow other mechanisms to remain operative and produce disease. Thus, a more accurate interpretation of the cited experiments is that the PrP^C^ N-terminal domain is not required for prion-induced neurodegeneration, but this domain could nevertheless still play a contributing role.

[Editors' note: further revisions were requested prior to acceptance, as described below.]

*The manuscript has been improved but there are some remaining issues that need to be addressed before acceptance, as outlined below:*

*Please revise the text of the manuscript to provide possible explanations for unanticipated results, as described by Reviewer #1.*

*Reviewer #1:*

*The authors have posited explanations for two of the deficits noted in my review of the manuscript, namely 1) that residues 91-104 may contribute to current induction in deltaCR/delta51-90 and are absent in 1-31-EGFP and 1-58-EGFP, or possibly that EGFP does not position 1-31 or 1-58 NTD sequences at the proper distance from the membrane; and 2) that no antibody-induced currents are observed in N2A cells transfected with delta23-31 PrP because this construct may compete for antibody binding or suppress endogenous PrP expression. These are valid potential explanations (though of course they could actually be tested!), but they do not make their way in any shape or form into the revised text. The authors should explicitly discuss these issues in the manuscript. Otherwise, despite the public review forum, these points will likely escape the notice of most readers.*

We have now prepared a second revised version in which we have incorporated into the Results section answers to the two questions raised by reviewer #1. We hope that the manuscript will now be acceptable for publication.